# Personalized Federated Learning with Mixture of Models for Adaptive Prediction and Model Fine-Tuning

**Pouya M. Ghari**
University of California Irvine
pmollaeb@uci.edu

**Yanning Shen** *
University of California Irvine
yannings@uci.edu

## Abstract

Federated learning is renowned for its efficacy in distributed model training, ensuring that users, called clients, retain data privacy by not disclosing their data to the central server that orchestrates collaborations. Most previous work on federated learning assumes that clients possess static batches of training data. However, clients may also need to make real-time predictions on streaming data in non-stationary environments. In such dynamic environments, employing pre-trained models may be inefficient, as they struggle to adapt to the constantly evolving data streams. To address this challenge, clients can fine-tune models online, leveraging their observed data to enhance performance. Despite the potential benefits of client participation in federated online model fine-tuning, existing analyses have not conclusively demonstrated its superiority over local model fine-tuning. To bridge this gap, the present paper develops a novel personalized federated learning algorithm, wherein each client constructs a personalized model by combining a locally fine-tuned model with multiple federated models learned by the server over time. Theoretical analysis and experiments on real datasets corroborate the effectiveness of this approach for real-time predictions and federated model fine-tuning.

## 1 Introduction

Federated learning enables a group of learners, known as *clients*, to collaborate and collectively train a model under the coordination of a central *server*, without revealing their data. In this framework, clients perform local model updates and share these updates with the server. By aggregating these local updates, the server globally updates the model. Many prior works in the literature have assumed that each client stores a batch of training data and updates models locally based on this stored data (see e.g., [38, 29, 47, 5, 34, 53]). However, in some cases, clients may need to make real-time predictions, and streams of data arrive sequentially, making it challenging to store and process data in batch. Furthermore, if clients operate in a non-stationary and dynamic environment, employing pre-trained models may fall short in prediction accuracy, requiring clients to fine-tune their models to adapt to their data.

A federated learning framework, commonly referred to as online federated learning [37, 26, 19], specifically addresses situations where clients engage in real-time predictions using a shared global model. After making predictions, clients collaborate with the server to update the global model for subsequent predictions in the future. Specifically, after making predictions, clients incur losses and based on the incurred losses, clients update the model locally and send their local updates to the server. The server updates the global model upon aggregating local updates and then distributes the updated global model to clients for use in the ongoing online prediction task. In this context, the

---

*Corresponding author

38th Conference on Neural Information Processing Systems (NeurIPS 2024).

performance of clients can be assessed using the notion of *regret* [4, 3]. The regret of a client at a given time step is defined as the difference between its prediction loss and that of the best model in hindsight. The best model in hindsight is determined as the model that achieves the minimum total prediction loss over time across all clients' data. The primary objective is to minimize the cumulative regret of all clients over time.

In the literature, federated learning algorithms based on online gradient descent have been proposed that achieve sub-linear regret upper bounds [37, 26, 19]. This suggests that over the long run, these algorithms perform as well as the best model in hindsight. However, in Section 3, this paper demonstrates that these federated learning algorithms do not obtain tighter regret bounds compared to the scenario where each client learns its own model locally without participating in federated learning. This indicates that participation in federated learning may not provide any benefit for clients performing online prediction. Specifically, if data is distributed non-i.i.d. among clients and data distributions are time-variant and not known a priori, it is likely that local model training by clients will achieve better prediction accuracy than participation in federated learning. Although the benefit of federated learning in online prediction is not evident in existing theoretical analyses, models learned through federated learning enjoy higher generalizability as they are trained on all data samples distributed among clients. This motivates the idea that combining the model learned through federated learning with the locally learned one may prove effective in scenarios where clients need to perform online prediction.

This paper proposes the Fed-POE (Federated Learning with Personalized Online Ensemble) algorithm, through which each client constructs a personalized model for online prediction by adaptively ensembling the locally learned model and the model learned through federated learning. With Fed-POE, clients can adapt their local models to their data while benefiting from the higher generalizability of the federated model. Theoretical analysis for convex cases demonstrates that Fed-POE achieves sublinear regret bounds with respect to both the best federated and local models in hindsight. This indicates that Fed-POE effectively leverages the advantages of both federated and local models. Providing such theoretical guarantees may not be feasible for non-convex models such as neural networks. These models may suffer from the forgetting process [51, 45], where fine-tuning on streaming data 'on the fly' can lead to forgetting previously observed data samples. To overcome this challenge, the present paper proposes a novel framework in which the server periodically stores federated model parameters over time. Each client adaptively selects a personalized subset of stored models on the server based on the performance of these models in the client's online prediction task. Clients then use the selected models, along with the federated and local models, to construct an ensemble model for prediction. Clients select a subset of models to both control the memory and computational complexity of prediction and to prune models with relatively lower accuracy, thereby improving prediction performance. Theoretical analysis proves that Fed-POE achieves sublinear regret with respect to the best model in hindsight among the local model, federated model, and all models stored by the server. The contributions of the present paper are summarized as follows:

- Fed-POE enables clients to utilize the advantages of both local and federated models for online prediction tasks.
- To address the issue of forgetting in online prediction, Fed-POE introduces a novel federated framework for collaboration between clients and the server.
- Theoretical analyses for both convex and non-convex cases prove that Fed-POE achieves sublinear regret with respect to the best model in hindsight.
- Extensive experiments on regression and image classification datasets show that Fed-POE effectively leverages the benefits of local and federated models, achieving higher online prediction accuracy compared to state-of-the-art federated learning algorithms.

## 2 Related Works

**Personalized Federated Learning.** Personalized federated learning involves developing individualized models for each client, derived from a global model learned through federated learning. Several personalized federated learning algorithms have been proposed in the literature [50, 6, 54, 56]. In [24, 11, 30, 33], clients construct their personalized models by adding a regularization term to the local objective and using the global federated model. Algorithms in [12, 1] allow clients to fine-tune the global federated model using model-agnostic meta-learning [14] to learn their personalized

models. With Fed-Rep [8], each client generates a representation using the global model and learns its own local head for prediction. Algorithms in [10, 32, 57] enable clients to achieve personalized models by combining local and global models. However, none of these works have addressed the problem of online prediction while clients collaborate on training their personalized models.

**Federated Learning with Streaming Data.** Several studies have explored the problem of federated learning when clients receive a stream of data in real time. While [7, 36, 35] investigate federated learning scenarios where clients receive new training data in each learning round, they do not address the aspect of online prediction by clients. Consequently, these works cannot provide regret guarantees for online prediction. Additionally, [9] studies the issue of staleness in online federated learning to both train and make prediction with the model; however, it lacks rigorous theoretical analysis. In [37], an online federated learning algorithm is introduced, utilizing online mirror descent, with a proven sublinear regret bound. Similarly, [42, 16] propose online federated learning algorithms with guaranteed sublinear regret. Furthermore, [43] analyzes the benefits of collaboration in online federated learning, particularly in scenarios where only loss values at queried points are available to clients, without access to loss gradients. Regarding online federated kernel learning, algorithms are introduced in [17, 23], albeit without accompanying regret analysis, leading to an absence of guaranteed regret bounds. In [26, 19], multiple kernel-based models and random feature-based online federated kernel learning algorithms are proposed, with demonstrated sublinear regret.

## 3 Preliminaries

This section explains the problem of federated learning for real-time prediction and model training. The present section studies the cases where clients either collaborate in federated learning or employ online gradient descent methods to locally train the model in real-time. This study highlights the motivation behind the proposed ensemble approach to federated learning.

### 3.1 Online Prediction and Federated Learning

Let there are $N$ clients interact with a server to train a model $f(\cdot; \cdot)$. Also, let $[N] := \{1, \ldots, N\}$. At each time step $t$, client $i$, $\forall i \in [N]$ receives a data sample $\boldsymbol{x}_{i,t} \in \mathbb{R}^d$ and makes the prediction $f(\boldsymbol{x}_{i,t}; \boldsymbol{\theta}_t)$ where $\boldsymbol{\theta}_t$ denotes the parameter of the model at time step $t$. Note that generalization to the scenario where at each time step, each client receives a dataset instead of a single data sample is straightforward. After making prediction, client $i$, $\forall i \in [N]$ observes label $y_{i,t}$. Then client $i$, $\forall i \in [N]$ computes the loss of its prediction $\mathcal{L}(f(\boldsymbol{x}_{i,t}; \boldsymbol{\theta}_t), y_{i,t})$ where $\mathcal{L}(\cdot, \cdot)$ denotes the loss function. After computing the loss, clients send their updates to the server (e.g., by sending the gradient loss $\nabla \mathcal{L}(f(\boldsymbol{x}_{i,t}; \boldsymbol{\theta}_t), y_{i,t})$) and the server aggregates information from clients to update $\boldsymbol{\theta}_t$ to $\boldsymbol{\theta}_{t+1}$ to be used by clients to make predictions at time step $t + 1$. For ease of presentation, it is assumed that all clients can send their updates to the server every time step. Generalizing the results for the cases where only a fraction of clients can send their updates is straightforward. Furthermore, it is assumed that the label $y_{i,t}$ for any $i$ and $t$ is determined by the environment through a process unknown to the clients. This implies that the data distribution observed by client $i$ can be non-stationary, and client $i$, $\forall i \in [N]$ does not know the distribution. Additionally, the data distribution can differ across clients. The goal is to enable clients to collaborate with the server to minimize the cumulative regret of clients over time. The regret of client $i$ at time step $t$ is defined as the difference between the prediction loss $\mathcal{L}(f(\boldsymbol{x}_{i,t}; \boldsymbol{\theta}_t), y_{i,t})$ and the loss $\mathcal{L}(f(\boldsymbol{x}_{i,t}; \boldsymbol{\theta}^*), y_{i,t})$ corresponding to the model with the optimal parameter $\boldsymbol{\theta}^*$. Therefore, the average cumulative regret of clients up to time horizon $T$ is defined as

$$R_T = \frac{1}{N} \sum_{t=1}^{T} \sum_{i=1}^{N} \mathcal{L}(f(\boldsymbol{x}_{i,t}; \boldsymbol{\theta}_t), y_{i,t}) - \frac{1}{N} \sum_{t=1}^{T} \sum_{i=1}^{N} \mathcal{L}(f(\boldsymbol{x}_{i,t}; \boldsymbol{\theta}^*), y_{i,t}) \tag{1}$$

where $\boldsymbol{\theta}^*$ denotes the optimal model parameter in hindsight and can be expressed as

$$\boldsymbol{\theta}^* = \arg\min_{\boldsymbol{\theta}} \sum_{t=1}^{T} \sum_{i=1}^{N} \mathcal{L}(f(\boldsymbol{x}_{i,t}; \boldsymbol{\theta}), y_{i,t}). \tag{2}$$

One common way to solve the problem is that clients employ online gradient descent to update models locally and the server exploits federated averaging [31, 37] to update the global model.

## 3.2 Federated Learning with Online Gradient Descent

At each time step $t$, client $i$ obtains the locally updated model parameter $\psi_{i,t+1}$ as follows:

$$\psi_{i,t+1} = \theta_t - \eta \nabla \mathcal{L}(f(x_{i,t}; \theta_t), y_{i,t}) \tag{3}$$

where $\eta$ is the learning rate. Aggregating locally updated model parameters, the server obtains the updated global model parameter for time step $t + 1$ as

$$\theta_{t+1} = \frac{1}{N} \sum_{i=1}^{N} \psi_{i,t+1}. \tag{4}$$

This continues up until the time horizon $T$. This paper examines regret under some or all of the following assumptions:

**A1.** The loss function $\mathcal{L}(f(x; \theta), y)$ is convex with respect to $\theta$.

**A2.** The gradient of the loss is bounded as $\|\nabla \mathcal{L}(f(x; \theta), y)\| \leq G$.

**A3.** For any $x$, $\theta$, the loss is bounded as $0 \leq \mathcal{L}(f(x; \theta), y) \leq 1$.

The following theorem specifies the regret bound for federated learning employing online gradient descent under A1 and A2.

**Theorem 1.** *Employing online gradient descent, the following cumulative regret upper bound is guaranteed for federated learning under assumptions A1 and A2:*

$$\frac{1}{N} \sum_{t=1}^{T} \sum_{i=1}^{N} \mathcal{L}(f(x_{i,t}; \theta_t), y_{i,t}) - \frac{1}{N} \sum_{t=1}^{T} \sum_{i=1}^{N} \mathcal{L}(f(x_{i,t}; \theta^*), y_{i,t}) \leq \frac{\|\theta^*\|^2}{2\eta} + \frac{\eta}{2} G^2 T. \tag{5}$$

Proof of Theorem 1 can be found in Appendix A. According to Theorem 1, choosing $\eta = \mathcal{O}(\sqrt{1/T})$, the cumulative regret in (5) is bounded from above as $\mathcal{O}(\sqrt{T})$. If the time horizon $T$ is unknown, the doubling trick technique (see e.g., [2, 22]) can be effectively used to set the learning rate to maintain theoretical guarantees. Consider the case where each client learns the model locally using online gradient descent without collaborating with other clients and the server. Let $\phi_{i,t}$ denote the local model parameter learned by client $i$ at time step $t$, employing online gradient descent locally. The regret of client $i$ in this case is equivalent to federated learning regret where there is only one client. Therefore, substituting $N = 1$ and $\theta_0 = 0$ in (27) of Appendix A, for the cumulative regret of client $i$ with respect to any $\theta$, we get

$$\sum_{t=1}^{T} \mathcal{L}(f(x_{i,t}; \phi_{i,t}), y_{i,t}) - \sum_{t=1}^{T} \mathcal{L}(f(x_{i,t}; \theta), y_{i,t}) \leq \frac{\|\theta\|^2}{2\eta} + \frac{\eta}{2} G^2 T. \tag{6}$$

Averaging (6) over all clients and substituting $\theta$ with $\theta^*$ in (6), we obtain

$$\frac{1}{N} \sum_{t=1}^{T} \sum_{i=1}^{N} \mathcal{L}(f(x_{i,t}; \phi_{i,t}), y_{i,t}) - \frac{1}{N} \sum_{t=1}^{T} \sum_{i=1}^{N} \mathcal{L}(f(x_{i,t}; \theta^*), y_{i,t}) \leq \frac{\|\theta^*\|^2}{2\eta} + \frac{\eta}{2} G^2 T. \tag{7}$$

Comparing (5) with (7), it can be inferred that under assumptions A1 and A2, federated learning does not achieve tighter regret bound than the case where each client independently learns its local model. Hence, from a theoretical standpoint, it remains uncertain whether collaboration in federated learning yields any improvement over local online model training. Intuitively, collaboration in federated learning may prove advantageous when there exists similarity among data samples observed by clients over time. However, in cases where data distribution is heterogeneous and such similarities are lacking, employing online local training may yield superior results for a client. Given the lack of prior information on data distribution in online scenarios, each client can independently assess over time whether utilizing the model learned through federated learning for predictions is beneficial.

Furthermore, according to assumption A1, the theoretical guarantees obtained in (5) and (7) hold if the loss is convex with respect to the model parameter. However, in the case of non-convex models, such as neural networks, these theoretical guarantees are not applicable. Non-convex models, like neural networks, may encounter the forgetting process [51, 45], where the model tends to overfit to recently observed data samples. Consequently, it may not be feasible to derive a single model parameter using online gradient descent that achieves sublinear regret with respect to the best model in hindsight. The subsequent section introduces a novel algorithm aimed at assisting clients in addressing such scenarios.

# 4 Personalized Federated Learning Methods

The current section introduces personalized federated learning algorithms where each client dynamically learns the prediction performance of both the models trained through federated learning over time and the locally learned model.

## 4.1 Ensemble Learning

At each time step, each client constructs an ensemble model comprising the federated model and its locally learned model to make a prediction. Let $\phi_{i,t}$ be the model parameter locally learned by client $i$ such that at each time step $t$, client $i$ updates $\phi_{i,t}$ via gradient descent as

$$\phi_{i,t+1} = \phi_{i,t} - \eta \nabla \mathcal{L}(f(\boldsymbol{x}_{i,t}; \phi_{i,t}), y_{i,t}). \tag{8}$$

Furthermore, let clients and the server collaborate to update the *federated* model parameter $\boldsymbol{\theta}_t$ as outlined in (3) and (4). At time step $t$, client $i$ makes the prediction for $\boldsymbol{x}_{i,t}$ using its personalized ensemble model $f_{i,t}(\cdot)$ expressed as

$$f_{i,t}(\boldsymbol{x}_{i,t}) = \frac{\alpha_{i,t}}{\alpha_{i,t} + \beta_{i,t}} f(\boldsymbol{x}_{i,t}; \boldsymbol{\theta}_t) + \frac{\beta_{i,t}}{\alpha_{i,t} + \beta_{i,t}} f(\boldsymbol{x}_{i,t}; \phi_{i,t}) \tag{9}$$

where $\alpha_{i,t}$ and $\beta_{i,t}$ represent the weights assigned by client $i$ to the federated and local models, respectively, indicating the credibility of predictions from each model. After making predictions and observing the label $y_{i,t}$, client $i$ computes the loss of predictions from the federated and local models, updating the weights $\alpha_{i,t}$ and $\beta_{i,t}$ using the multiplicative update rule as follows:

$$\alpha_{i,t+1} = \alpha_{i,t} \exp\left(-\eta_c \mathcal{L}(f(\boldsymbol{x}_{i,t}; \boldsymbol{\theta}_t), y_{i,t})\right), \tag{10a}$$
$$\beta_{i,t+1} = \beta_{i,t} \exp\left(-\eta_c \mathcal{L}(f(\boldsymbol{x}_{i,t}; \phi_{i,t}), y_{i,t})\right) \tag{10b}$$

where $\eta_c$ is a learning rate. Client $i$ initializes $\alpha_{i,1} = 1$ and $\beta_{i,1} = 1$. The proposed algorithm is personalized since according to (9), each client constructs its own ensemble model to perform prediction. The personalized regret of client $i$ is defined as $C_{i,T} = \sum_{t=1}^{T} \mathcal{L}(f_{i,t}(\boldsymbol{x}_{i,t}), y_{i,t}) - \sum_{t=1}^{T} \mathcal{L}(f(\boldsymbol{x}_{i,t}; \phi_i^*), y_{i,t})$ where $\phi_i^*$ denotes the best hindsight model parameter for client $i$ which can be expressed as $\phi_i^* = \arg\min_{\phi} \sum_{t=1}^{T} \mathcal{L}(f(\boldsymbol{x}_{i,t}; \phi_i), y_{i,t})$. The following theorem establishes the personalized regret upper bound for client $i$ as well as global regret of all clients with respect to the best model parameter in hindsight.

**Theorem 2.** *Under assumptions A1–A3, employing the ensemble model in* (9) *for online prediction, the global regret of all clients is bounded from above as*

$$\frac{1}{N} \sum_{t=1}^{T} \sum_{i=1}^{N} \mathcal{L}(f_{i,t}(\boldsymbol{x}_{i,t}), y_{i,t}) - \frac{1}{N} \sum_{t=1}^{T} \sum_{i=1}^{N} \mathcal{L}(f(\boldsymbol{x}_{i,t}; \boldsymbol{\theta}^*), y_{i,t}) \leq \frac{\|\boldsymbol{\theta}^*\|^2}{2\eta} + \frac{\ln(2)}{\eta_c} + \frac{\eta}{2} G^2 T + \frac{\eta_c T}{2} \tag{11}$$

*while client $i$ achieves the following personalized regret upper bound:*

$$\sum_{t=1}^{T} \mathcal{L}(f_{i,t}(\boldsymbol{x}_{i,t}), y_{i,t}) - \sum_{t=1}^{T} \mathcal{L}(f(\boldsymbol{x}_{i,t}; \phi_i^*), y_{i,t}) \leq \frac{\|\phi_i^*\|^2}{2\eta} + \frac{\ln(2)}{\eta_c} + \frac{\eta}{2} G^2 T + \frac{\eta_c T}{2}. \tag{12}$$

Proof of Theorem 2 is given in Appendix B. According to (11) and (12) in Theorem 2, setting $\eta = \mathcal{O}(1/\sqrt{T})$, $\eta_c = \mathcal{O}(1/\sqrt{T})$ both the global regret for all clients and the personalized regret of each client $i$ achieve a regret bound of $\mathcal{O}(\sqrt{T})$. This demonstrates that while the ensemble model in (9) ensures personalized regret guarantees for each client with respect to its best model in hindsight, it also enables clients to leverage federated learning, thus enjoying sublinear regret in comparison to the best global model in hindsight.

**Comparison with Federated and Local Models.** The main advantage of using the ensemble model instead of the federated model lies in Theorem 2, where it is shown that the ensemble model can attain the global regret guarantee provided by the federated model while employing the federated model, achieving the personalized regret guarantee in (12) is not feasible. However, according to (6) and (7), using the online local training, each client achieves sublinear regret with respect to its best model while all clients achieve sublinear regret with respect to the best global model in hindsight.

---

**Algorithm 1** Model selection by client $i$ at time step $t$

---

1: **Input:** $\mathbb{D}_t$, $M$, wights $\{w_{ij,t}\}_{j=1}^{|\mathbb{D}_t|}$ and $\mathbb{M}_{i,t} = \varnothing$.
2: **for** $m = 1, \ldots, M$ **do**
3:     Sample model index $k$ according to the PMF $p_{ij,t} = \frac{w_{ij,t}}{\sum_{j=1}^{|\mathbb{D}_t|} w_{ij,t}}, \forall j \in [|\mathbb{D}_t|]$.
4:     **if** $k \notin \mathbb{M}_{i,t}$ **then**
5:         Append model index $k$ to $\mathbb{M}_{i,t}$.
6:     **end if**
7: **end for**
8: **Output:** $\mathbb{M}_{i,t}$.

---

This efficacy of local training stems from clients adapting the model to their individual data. In contrast, in federated learning, the model is trained on all data samples across clients, potentially leading to higher generalizability compared to its local counterpart. If there are similarities in the distribution of data samples among clients over time, the federated model is anticipated to achieve greater accuracy in online prediction. However, due to the lack of available information regarding the relationships between data samples observed by clients before online prediction, these advantages may not be reflected in theoretical bounds. The proposed method constructs an ensemble to harness the advantages of both federated and local models for online prediction, as evidenced by Theorem 2. Experimental results in Subsection in 5.1 confirm that the ensemble model achieves superior performance compared to both local and federated models.

### 4.2 Model Selection

Regret guarantees for the ensemble method, as outlined in Theorem 2, are contingent upon the model's convexity. However, if the model is non-convex, achieving such guarantees may not be feasible. Particularly, non-convex models such as neural networks are susceptible to the forgetting process [51, 45], wherein applying online gradient descent may lead to overfitting to recently observed data samples. This section introduces a novel algorithm that allows clients to make online predictions using non-convex models while simultaneously collaborating to fine-tune the model. The scenario assumes the existence of a pre-trained model, and the objective for clients is two-fold: to make real-time predictions and to refine the model for alignment with their preferences. This situation may arise, for instance, in fine-tuning large foundation models to tailor them to client preferences.

Let the server and clients collaborate to fine-tune the non-convex model $f(\cdot; \cdot)$. At each time step $t$, client $i$ updates the model on the batch of recently observed samples with size $b$ as

$$\psi_{i,t+1} = \theta_t - \frac{\eta}{b} \sum_{\tau=t-b}^{t} \nabla \mathcal{L}(f(\boldsymbol{x}_{i,\tau}; \boldsymbol{\theta}_t), y_{i,\tau}). \tag{13}$$

Then the server aggregates locally updated parameters and updates the federated model parameter as in (4). Furthermore, each clients learns its own local model by fine-tuning the pre-trained model locally via online gradient descent as in (8) on the batch of $b$ recently observed samples. While online gradient descent methods are well-known for their efficiency in handling dynamic environments, employing the update rule of (13) for non-convex models may lead to overfitting to recently observed batches. To mitigate potential forgetting, the server saves the federated model parameters every $n$ time step, where $n$ is an integer hyperparameter.

Let $\mathbb{D}_t$ represent the set of model parameters stored by the server at time step $t$. At time step $\tau = (j-1)n + 1$, the server adds $\boldsymbol{\theta}_\tau$ to $\mathbb{D}_\tau$ meaning that $\mathbb{D}_{\tau+1} = \mathbb{D}_\tau \cup \{\boldsymbol{\theta}_\tau\}$. Let $\boldsymbol{\rho}_j$ denotes the $j$-th model parameter in $\mathbb{D}_t$. It can be concluded that $\boldsymbol{\rho}_j = \boldsymbol{\theta}_{(j-1)n+1}$. The server continues saving model parameters every $n$ time steps until time step $U \leq T$. Client $i$ assigns a weight $w_{ij,t}$ to the $j$-th model in $\mathbb{D}_t$, which assesses the credibility of the predictions given by the model parameter $\boldsymbol{\rho}_j$. Client $i$ initializes $w_{ij,1} = 1$. At each time step $t$, client $i$ selects $M$ models with replacement from $\mathbb{D}_t$ to construct its model set $\mathbb{M}_{i,t}$. Algorithm 1 illustrates the model selection process conducted by client $i$ at time step $t$. During each round of model selection, client $i$ chooses a model according to a probability mass function (PMF) proportional to weights $\{w_{ij,t} | 1 \leq j \leq |\mathbb{D}_t|\}$ where $|\cdot|$ denotes the cardinality of a set (see step 3 in Algorithm 1). Client $i$ adds the chosen model to $\mathbb{M}_{i,t}$ if it is not already present. Therefore, it can be concluded that $|\mathbb{M}_{i,t}| \leq M, \forall i, t$. Then at each time step $t$,

---

**Algorithm 2** Fed-POE: Federated Learning with Personalized Online Ensemble

---
1: **Input:** Model $f(\cdot;\cdot)$, batch size $b$, $n$, $U$, $\mathbb{D}_{i,1} = \varnothing$.
2: **for** $t = 1, \ldots, T$ **do**
3:     The server transmits $\boldsymbol{\theta}_t$ to all clients.
4:     **for all** $i \in [N]$, client $i$ **do**
5:         Performs model selection given $\mathbb{D}_t$ according to Algorithm 1 to obtain $\mathbb{M}_{i,t}$.
6:         Makes prediction $\bar{f}_{i,t}(\boldsymbol{x}_{i,t})$ as in (15) using chosen model set $\mathbb{M}_{i,t}$.
7:         Upon receiving $y_{i,t}$, updates $\alpha_{i,t}, \beta_{i,t}, \gamma_{i,t}, \delta_{i,t}$ and $\{w_{ij,t}\}_{j=1}^{|\mathbb{D}_t|}$ as in (10), (16) and (17).
8:         Updates the local model as $\boldsymbol{\phi}_{i,t+1} = \boldsymbol{\phi}_{i,t} - \frac{\eta}{b}\sum_{\tau=t-b}^{t} \nabla\mathcal{L}(f(\boldsymbol{x}_{i,t}; \boldsymbol{\phi}_{i,\tau}), y_{i,t})$.
9:         Updates the federated model as $\boldsymbol{\psi}_{i,t+1} = \boldsymbol{\theta}_t - \frac{\eta}{b}\sum_{\tau=t-b}^{t} \nabla\mathcal{L}(f(\boldsymbol{x}_{i,\tau}; \boldsymbol{\theta}_t), y_{i,\tau})$.
10:        Sends $\boldsymbol{\psi}_{i,t+1}$ to the server.
11:     **end for**
12:     **if** $t \leq U$ and $t \bmod n = 0$ **then**
13:         The server updates $\mathbb{D}_t$ as $\mathbb{D}_{t+1} = \mathbb{D}_t \cup \{\boldsymbol{\theta}_t\}$.
14:     **end if**
15:     The server updates $\boldsymbol{\theta}_t$ as $\boldsymbol{\theta}_{t+1} = \frac{1}{N}\sum_{i=1}^{N} \boldsymbol{\psi}_{i,t+1}$.
16: **end for**

---

client $i$ downloads all models in $\mathbb{M}_{i,t}$ from the server. Upon receiving the models, client $i$ constructs an ensemble model $\tilde{f}_{i,t}(\cdot)$ as

$$\tilde{f}_{i,t}(\boldsymbol{x}) = \sum_{j \in \mathbb{M}_{i,t}} \frac{w_{ij,t}}{\sum_{m \in \mathbb{M}_{i,t}} w_{im,t}} f(\boldsymbol{x}; \boldsymbol{\rho}_j). \tag{14}$$

Client $i$ makes the prediction for $\boldsymbol{x}_{i,t}$ as

$$\bar{f}_{i,t}(\boldsymbol{x}_{i,t}) = \frac{\gamma_{i,t}}{\gamma_{i,t} + \delta_{i,t}} f_{i,t}(\boldsymbol{x}_{i,t}) + \frac{\delta_{i,t}}{\gamma_{i,t} + \delta_{i,t}} \tilde{f}_{i,t}(\boldsymbol{x}_{i,t}) \tag{15}$$

where $f_{i,t}(\boldsymbol{x}_{i,t})$ is the ensemble of local and federated models as defined in (9). Furthermore, $\gamma_{i,t}$ and $\delta_{i,t}$ are weights assigned by client $i$ to ensemble models $f_{i,t}(\cdot)$ and $\tilde{f}_{i,t}(\cdot)$, respectively. Upon observing the label $y_{i,t}$ after prediction, client $i$ updates weights $\gamma_{i,t}$ and $\delta_{i,t}$ as

$$\gamma_{i,t+1} = \gamma_{i,t} \exp\left(-\eta_c \mathcal{L}(f_{i,t}(\boldsymbol{x}_{i,t}), y_{i,t})\right), \tag{16a}$$

$$\delta_{i,t+1} = \delta_{i,t} \exp(-\eta_c \mathcal{L}(\tilde{f}_{i,t}(\boldsymbol{x}_{i,t}), y_{i,t})). \tag{16b}$$

Furthermore, $\alpha_{i,t}$ and $\beta_{i,t}$ which are used to construct $f_{i,t}(\boldsymbol{x}_{i,t})$ in (9) are updated as in (10). The weight $w_{ij,t}$ is updated using the importance sampling loss as

$$w_{ij,t+1} = w_{ij,t} \exp\left(-\eta_c \frac{\mathcal{L}(f(\boldsymbol{x}_{i,t}; \boldsymbol{\rho}_j), y_{i,t})}{q_{ij,t}} \mathbf{1}_{j \in \mathbb{M}_{i,t}}\right) \tag{17}$$

where $\mathbf{1}_{j \in \mathbb{M}_{i,t}}$ denotes the indicator function and is 1 if $j \in \mathbb{M}_{i,t}$. Moreover, $q_{ij,t}$ is the probability that client $i$ selects the $j$-th model in $\mathbb{D}_t$ to be in $\mathbb{M}_{i,t}$ and can be expressed as $q_{ij,t} = 1 - (1 - p_{ij,t})^M$ where $p_{ij,t}$ is defined in step 3 of Algorithm 1. The proposed algorithm, named Federated Learning with Personalized Online Ensemble (Fed-POE) is summarized in Algorithm 2. It is useful to note that the proposed method in Subsection 4.1 is a special case of Fed-POE by setting $M = 0$ and $b = 1$.

**Efficiency of Fed-POE.** To perform model selection using Fed-POE, clients do not need to store all model parameters in $\mathbb{D}_t$ for all $t \in [T]$. Instead, the server, which has higher storage capacity than the clients, stores all model parameters, and clients download a subset of at most $M$ model parameters. The hyperparameter $M$ can be chosen such that clients can handle the memory and computational requirements of making predictions with the selected subset of models. Let $C_F$ denote the number of computations required to fine-tune the model $f$, and let $C_I$ represent the number of computations required to make an inference with model $f$. Assume that the complexity of model selection in Algorithm 1 is negligible compared to fine-tuning and making inferences with model $f$. According to Algorithm 2, each client performs $2C_F + (M + 2)C_I$ computations per time step. Therefore, the computational complexity of Fed-POE for each client is $\mathcal{O}(C_F + MC_I)$. Beyond memory and computational considerations, selecting a subset of models from $\mathbb{D}_t$ helps clients improve their

prediction accuracy. Specifically, using the model weights $\{w_{ij,t}\}$ at time step $t$, client $i$ selects models that perform better on its data while pruning those with lower performance.

Let $h_j(\boldsymbol{x}_{i,t}) = f(\boldsymbol{x}_{i,t}; \boldsymbol{\rho}_j)$ denote the model associated with the $j$-th model parameter in $\mathbb{D}_t$. Furthermore, $h_{\text{loc}}(\boldsymbol{x}_{i,t}) = f(\boldsymbol{x}_{i,t}; \boldsymbol{\phi}_{i,t})$ and $h_{\text{fed}}(\boldsymbol{x}_{i,t}) = f(\boldsymbol{x}_{i,t}; \boldsymbol{\theta}_t)$ represent the local and federated models, respectively, . Let the set of models $\mathbb{H}$ be defined as $\mathbb{H} := \{h_j \mid \forall j : 1 \leq j \leq |\mathbb{D}_T|\} \cup \{h_{\text{loc}}, h_{\text{fed}}\}$. This set $\mathbb{H}$ includes all models that can be employed by each client using Fed-POE. The best model in hindsight $h^*$ and the best model in hindsight for client $i$, $h_i^*$ are defined as

$$h^* = \min_{h \in \mathbb{H}} \sum_{t=1}^{T} \sum_{i=1}^{N} \mathcal{L}(h(\boldsymbol{x}_{i,t}), y_{i,t}), \tag{18a}$$

$$h_i^* = \min_{h \in \mathbb{H}} \sum_{t=1}^{T} \mathcal{L}(h(\boldsymbol{x}_{i,t}), y_{i,t}). \tag{18b}$$

The following theorem provides the regret upper bound of Fed-POE.

**Theorem 3.** *Under assumption A3, employing Fed-POE in Algorithm 2, the expected global regret of all clients is bounded from above as*

$$\frac{1}{N} \sum_{t=1}^{T} \sum_{i=1}^{N} \mathbb{E}_t[\mathcal{L}(\bar{f}_{i,t}(\boldsymbol{x}_{i,t}), y_{i,t})] - \frac{1}{N} \sum_{t=1}^{T} \sum_{i=1}^{N} \mathcal{L}(h^*(\boldsymbol{x}_{i,t}), y_{i,t})$$
$$\leq \frac{\ln 2U - \ln 2n}{\eta_c} + \frac{\eta_c}{2}(\frac{U}{n} + 1)T + (1 - \frac{\eta_c}{2n}U)U \tag{19}$$

*while client $i$ achieves the following expected personalized regret upper bound:*

$$\sum_{t=1}^{T} \mathbb{E}_t[\mathcal{L}(\bar{f}_{i,t}(\boldsymbol{x}_{i,t}), y_{i,t})] - \sum_{t=1}^{T} \mathcal{L}(h_i^*(\boldsymbol{x}_{i,t}), y_{i,t})$$
$$\leq \frac{\ln 2U - \ln 2n}{\eta_c} + \frac{\eta_c}{2}(\frac{U}{n} + 1)T + (1 - \frac{\eta_c}{2n}U)U. \tag{20}$$

*The expectation is taken with respect to randomization in model selection.*

The proof of Theorem 3 can be found in Appendix C. According to (19) and (20), setting $\eta_c = \mathcal{O}(1/\sqrt{T})$, $n = \mathcal{O}(\sqrt{T})$ and $U = \mathcal{O}(\sqrt{T})$, both the personalized and global regrets of clients achieve sublinear regret of $\mathcal{O}(\sqrt{T})$. Since clients construct their ensemble models using a time-varying subset of models, employing existing model selection and ensemble learning approaches [13, 15, 39, 46, 20, 41, 21] may not ensure the sublinear regrets stated in Theorem 3. However, by using the proposed Algorithm 1, Fed-POE guarantees sublinear regret bounds while allowing clients the flexibility to select time-varying and personalized subsets of models for their ensembles.

## 5 Experiments

The present section studies the performance of Fed-POE in Algorithm 2 compared to other baselines. Experiments are conducted on both image classification and regression tasks. The performance of federated learning is examined in both convex and non-convex cases. Codes are available at `https://github.com/pouyamghari/Fed-POE`.

### 5.1 Regression

The performance of the proposed Fed-POE is evaluated on online regression tasks. For these tasks, clients and the server collaborate to train a random feature kernel-based model, which is known to be convex [25, 48, 18]. Details about the random feature kernel-based model used in the experiments can be found in Appendix D. The performance of Fed-POE is compared with a baseline called Local, where clients train their models locally without participating in federated learning. Additionally, Fed-POE is compared to personalized federated learning baselines Ditto [30] and Fed-Rep [8], the online federated learning baseline Fed-OMD [37], and online federated

kernel learning baselines eM-KOFL [26] and POF-MKL [19]. Mean square error (MSE) is used as the metric to evaluate the performance of algorithms on regression task. MSE for client $i$ can be expressed as $\text{MSE}_i = \frac{1}{T}\sum_{t=1}^{T}(\hat{y}_{ij,t} - y_{i,t})^2$ where $\hat{y}_{ij,t}$ denotes the prediction of client $i$ at time step $t$. The performance of algorithms are tested on two regression datasets Air [58] and WEC [40]. Air dataset is a time-series dataset that each data sample contains air quality features and the goal is to predict the concentration of contamination in the air. Each sample in WEC dataset contains features of different wave energy converters and the goal is to predict power output. Data samples are distributed non-i.i.d among $400$ clients. Time horizon $T$ for both datasets is 250. More information about datasets and distributed data among clients is presented in Appendix D.

Table 1 presents the MSE of algorithms and their standard deviation across clients. For all algorithms, the learning rates are set to $\eta = \eta_c = 1/\sqrt{T}$. Table 1 shows that when data is distributed non-i.i.d. among clients, local model training can achieve higher accuracy compared to federated learning. For the Air dataset, Local achieves lower MSE than other federated learning baselines except for Fed-POE. For the WEC dataset, only POF-MKL achieves lower MSE than Local. This indicates that the performance of federated learning approaches compared to Local depends on the dataset. By utilizing both federated and local models, Fed-POE achieves the lowest MSE. Table 1 shows that the performance of Fed-POE relative to other baselines is more consistent across different datasets.

Table 1: Average MSE ($\times 10^{-3}$) and its standard deviation ($\times 10^{-3}$) across clients for online regression on Air and WEC datasets.

| Methods | Air | WEC |
|---|---|---|
| Local | $9.12 \pm 3.59$ | $17.64 \pm 0.44$ |
| Ditto | $10.65 \pm 5.69$ | $33.88 \pm 16.08$ |
| Fed-Rep | $10.48 \pm 5.23$ | $35.13 \pm 10.38$ |
| Fed-OMD | $11.48 \pm 6.84$ | $32.61 \pm 27.38$ |
| eM-KOFL | $11.51 \pm 6.71$ | $72.29 \pm 62.48$ |
| POF-MKL | $10.66 \pm 6.07$ | $16.94 \pm 15.92$ |
| Fed-POE | $\mathbf{9.06 \pm 3.73}$ | $\mathbf{11.83 \pm 4.60}$ |

## 5.2 Image Classification

The performance of the proposed Fed-POE on an image classification task is compared with Local, Ditto [30], Fed-Rep [8], Fed-OMD [37], Fed-ALA [57], and Fed-DS [35]. Fed-ALA [57] is a personalized federated learning model suitable for deep neural networks, while Fed-DS [35] is a federated learning algorithm designed to handle data streams. Experiments are conducted on CIFAR-10 [28] and Fashion MNIST (FMNIST) [55] datasets. CIFAR-10 and FMNIST contain $60,000$ and $70,000$ images. For both CIFAR-10 and FMNIST, a CNN with a VGG architecture [49], consisting of two blocks, is pre-trained on a subset of training samples from each dataset. The training datasets are biased towards class 0. More details about training the CNNs can be found in Appendix D. For both the CIFAR-10 and FMNIST datasets, $10,000$ test samples are sequentially received by clients. There are 20 clients in total, and the data samples are distributed non-i.i.d. among them. For CIFAR-10, each client is biased towards one specific class, with $55\%$ of the samples belonging to that class and $5\%$ of the samples belonging to each of the other 9 classes. For FMNIST, each client is biased towards two classes, and the distribution of samples is time-variant. More details about the data distribution among clients and experimental setup can be found in Appendix D. At each time step, all algorithms employ batch of size 10 for model update. The learning rates for all algorithms are set to $\eta = 0.01/\sqrt{T}$ and $\eta_c = 1/\sqrt{T}$. The server stores models every $n = 20$ time steps. The metric to evaluate the performance of algorithms is the accuracy. The accuracy for client $i$ is defined as $\text{Accuracy}_i = \frac{1}{T}\sum_{t=1}^{T}\mathbf{1}_{\hat{y}_{i,t}=y_{i,t}}$ where $\hat{y}_{i,t}$ denotes the label predicted by client $i$ at time step $t$.

Average accuracy and its standard deviation across clients for CIFAR-10 and FMNIST are reported in Table 2. At each time step $t$, clients can store 10 model parameters. Therefore, $M$ is set to $M = 8$ for Fed-POE. The results indicate that the performance of Local relative to federated learning baselines depends on the dataset. While Local outperforms all federated baselines except for Fed-Rep and Fed-POE on FMNIST, it obtains the worst accuracy on CIFAR-10. Conversely, Fed-POE achieves the highest accuracy for both datasets, indicating

Table 2: Average accuracy and its standard deviation across clients for image classification.

| Methods | CIFAR-10 | FMNIST |
|---|---|---|
| Local | $50.35\% \pm 10.11\%$ | $78.81\% \pm 2.12\%$ |
| Ditto | $56.87\% \pm 9.06\%$ | $78.73\% \pm 1.89\%$ |
| Fed-Rep | $63.86\% \pm 7.97\%$ | $79.04\% \pm 1.77\%$ |
| Fed-OMD | $65.09\% \pm 7.39\%$ | $74.60\% \pm 6.52\%$ |
| Fed-ALA | $61.48\% \pm 8.88\%$ | $75.13\% \pm 6.39\%$ |
| Fed-DS | $64.26\% \pm 7.03\%$ | $75.62\% \pm 6.58\%$ |
| Fed-POE | $\mathbf{66.54\% \pm 8.07\%}$ | $\mathbf{79.23\% \pm 1.88\%}$ |

Table 3: Average accuracy and standard deviation across clients employing Fed-POE for image classification on CIFAR-10 with varying values of $M$ and batch size $b$.

| | $M = 0$ | $M = 4$ | $M = 8$ | $M = 16$ |
|---|---|---|---|---|
| $b = 1$ | $53.80\% \pm 6.71\%$ | $62.73\% \pm 8.29\%$ | $62.73\% \pm 8.29\%$ | $62.73\% \pm 8.26\%$ |
| $b = 10$ | $65.55\% \pm 8.77\%$ | $66.50\% \pm 8.00\%$ | $66.54\% \pm 8.08\%$ | $66.46\% \pm 7.98\%$ |
| $b = 20$ | $65.72\% \pm 8.62\%$ | $66.13\% \pm 8.20\%$ | $66.64\% \pm 7.94\%$ | $66.53\% \pm 8.00\%$ |
| $b = 30$ | $66.83\% \pm 8.54\%$ | $66.32\% \pm 7.92\%$ | $66.24\% \pm 8.05\%$ | $66.39\% \pm 8.02\%$ |

that Fed-POE efficiently leverages the advantages of both federated and local models. To analyze the effect of the number of models $M$ and batch size $b$ on the Fed-POE performance, experiments are conducted on the CIFAR-10 dataset, varying the batch size $b$ and the number of models $M$ selected by each client to construct the ensemble model. As observed in Table 3, the batch size $b = 1$ results in the worst accuracy, mainly due to the forgetting process where models overfit to the most recently observed data. However, increasing the batch size from $b = 10$ or $b = 20$ to $b = 30$ does not significantly improve the accuracy. Larger batch sizes may lead the model to perform better on older data, as the model is trained on older data over more iterations. Therefore, this study concludes that a moderate batch size is optimal, considering that increasing the batch size also increases computational complexity. Table 4 in Appendix D presents the accuracy of Fed-POE on both the CIFAR-10 and FMNIST datasets with varying values of $M$. The table shows that employing the saved models by the server in $\mathbb{D}_t$ improves performance, as setting $M = 0$ results in the worst accuracy. Moreover, it can be observed that increasing $M$ does not necessarily lead to further accuracy improvement. This aligns with the intuition behind selecting a subset of models rather than using all models.

## 6    Conclusions

This paper proposed Fed-POE, a personalized federated learning algorithm designed for online prediction and model fine-tuning. Fed-POE constructs an ensemble using local models and federated models stored by the server periodically over time. Theoretical analysis demonstrated that Fed-POE achieves sublinear regret. Experimental results revealed that the relative performance of local models compared to federated models depends on the dataset, making the decision between local model training and federated learning challenging. However, experimental results also show that Fed-POE consistently outperforms both local and federated models across all datasets. This indicates that Fed-POE effectively leverages the advantages of both local and federated models.

## Acknowledgement

Work in this paper is supported by NSF ECCS 2207457 and NSF ECCS 2412484.

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

# A  Proof of Theorem 1

This section provides the proof of Theorem 1. The proof follows similar steps to those in [37], and it is included here for the sake of completeness and to make the paper self-contained.

According to (3) and (4), for any $\boldsymbol{\theta}$ it can be written that

$$
\|\boldsymbol{\theta}_{t+1} - \boldsymbol{\theta}\|^2 = \left\| \frac{1}{N} \sum_{i=1}^{N} \boldsymbol{\phi}_{i,t+1} - \boldsymbol{\theta} \right\|^2 = \left\| \boldsymbol{\theta}_t - \frac{\eta}{N} \sum_{i=1}^{N} \nabla \mathcal{L}(f(\boldsymbol{x}_{i,t}; \boldsymbol{\theta}_t), y_{i,t}) - \boldsymbol{\theta} \right\|^2
$$

$$
= \|\boldsymbol{\theta}_t - \boldsymbol{\theta}\|^2 + \left\| \frac{\eta}{N} \sum_{i=1}^{N} \nabla \mathcal{L}(f(\boldsymbol{x}_{i,t}; \boldsymbol{\theta}_t), y_{i,t}) \right\|^2
$$

$$
- \frac{2\eta}{N} \sum_{i=1}^{N} (\boldsymbol{\theta}_t - \boldsymbol{\theta})^\top \nabla \mathcal{L}(f(\boldsymbol{x}_{i,t}; \boldsymbol{\theta}_t), y_{i,t}). \tag{21}
$$

Due to convexity of $\mathcal{L}(\cdot, \cdot)$ it can be concluded that

$$
\frac{2\eta}{N} \sum_{i=1}^{N} \mathcal{L}(f(\boldsymbol{x}_{i,t}; \boldsymbol{\theta}_t), y_{i,t}) - \mathcal{L}(f(\boldsymbol{x}_{i,t}; \boldsymbol{\theta}), y_{i,t}) \leq \frac{2\eta}{N} \sum_{i=1}^{N} (\boldsymbol{\theta}_t - \boldsymbol{\theta})^\top \nabla \mathcal{L}(f(\boldsymbol{x}_{i,t}; \boldsymbol{\theta}_t), y_{i,t}). \tag{22}
$$

Combining (21) with (22), we get

$$
\frac{2\eta}{N} \sum_{i=1}^{N} \mathcal{L}(f(\boldsymbol{x}_{i,t}; \boldsymbol{\theta}_t), y_{i,t}) - \frac{2\eta}{N} \sum_{i=1}^{N} \mathcal{L}(f(\boldsymbol{x}_{i,t}; \boldsymbol{\theta}), y_{i,t})
$$

$$
\leq \|\boldsymbol{\theta}_t - \boldsymbol{\theta}\|^2 - \|\boldsymbol{\theta}_{t+1} - \boldsymbol{\theta}\|^2 + \left\| \frac{\eta}{N} \sum_{i=1}^{N} \nabla \mathcal{L}(f(\boldsymbol{x}_{i,t}; \boldsymbol{\theta}_t), y_{i,t}) \right\|^2. \tag{23}
$$

Using assumption A2 and Arithmetic Mean Geometric Mean (AM-GM) inequality it can be written that

$$
\left\| \sum_{i=1}^{N} \nabla \mathcal{L}(f(\boldsymbol{x}_{i,t}; \boldsymbol{\theta}_t), y_{i,t}) \right\|^2 \leq N \sum_{i=1}^{N} \|\nabla \mathcal{L}(f(\boldsymbol{x}_{i,t}; \boldsymbol{\theta}_t), y_{i,t})\|^2 \leq N^2 G^2. \tag{24}
$$

Combining (23) with (24), we arrive at

$$
\frac{2\eta}{N} \sum_{i=1}^{N} \mathcal{L}(f(\boldsymbol{x}_{i,t}; \boldsymbol{\theta}_t), y_{i,t}) - \frac{2\eta}{N} \sum_{i=1}^{N} \mathcal{L}(f(\boldsymbol{x}_{i,t}; \boldsymbol{\theta}), y_{i,t}) \leq \|\boldsymbol{\theta}_t - \boldsymbol{\theta}\|^2 - \|\boldsymbol{\theta}_{t+1} - \boldsymbol{\theta}\|^2 + \eta^2 G^2. \tag{25}
$$

Dividing both sides by $\frac{2\eta}{N}$ yields

$$
\sum_{i=1}^{N} \mathcal{L}(f(\boldsymbol{x}_{i,t}; \boldsymbol{\theta}_t), y_{i,t}) - \sum_{i=1}^{N} \mathcal{L}(f(\boldsymbol{x}_{i,t}; \boldsymbol{\theta}), y_{i,t}) \leq \frac{N(\|\boldsymbol{\theta}_t - \boldsymbol{\theta}\|^2 - \|\boldsymbol{\theta}_{t+1} - \boldsymbol{\theta}\|^2)}{2\eta} + \frac{\eta N}{2} G^2. \tag{26}
$$

Summing (26) over time, we obtain

$$
\sum_{t=1}^{T} \sum_{i=1}^{N} \mathcal{L}(f(\boldsymbol{x}_{i,t}; \boldsymbol{\theta}_t), y_{i,t}) - \sum_{t=1}^{T} \sum_{i=1}^{N} \mathcal{L}(f(\boldsymbol{x}_{i,t}; \boldsymbol{\theta}), y_{i,t})
$$

$$
\leq \frac{N(\|\boldsymbol{\theta}_0 - \boldsymbol{\theta}\|^2 - \|\boldsymbol{\theta}_{T+1} - \boldsymbol{\theta}\|^2)}{2\eta} + \frac{\eta N}{2} G^2 T. \tag{27}
$$

Plugging in $\boldsymbol{\theta} = \boldsymbol{\theta}^*$ and $\boldsymbol{\theta}_0 = \boldsymbol{0}$ into (27) and considering the fact that $\|\boldsymbol{\theta}_{T+1} - \boldsymbol{\theta}\|^2 \geq 0$, we find

$$
\sum_{t=1}^{T} \sum_{i=1}^{N} \mathcal{L}(f(\boldsymbol{x}_{i,t}; \boldsymbol{\theta}_t), y_{i,t}) - \sum_{t=1}^{T} \sum_{i=1}^{N} \mathcal{L}(f(\boldsymbol{x}_{i,t}; \boldsymbol{\theta}^*), y_{i,t}) \leq \frac{N\|\boldsymbol{\theta}^*\|^2}{2\eta} + \frac{\eta N}{2} G^2 T \tag{28}
$$

which proves the Theorem.

## B Proof of Theorem 2

According to (10), we can write

$$\frac{\alpha_{i,t+1} + \beta_{i,t+1}}{\alpha_{i,t} + \beta_{i,t}} = \frac{\alpha_{i,t}}{\alpha_{i,t} + \beta_{i,t}} \exp\left(-\eta_c \mathcal{L}(f(\boldsymbol{x}_{i,t}; \boldsymbol{\theta}_t), y_{i,t})\right)$$
$$+ \frac{\beta_{i,t}}{\alpha_{i,t} + \beta_{i,t}} \exp\left(-\eta_c \mathcal{L}(f(\boldsymbol{x}_{i,t}; \boldsymbol{\phi}_{i,t}), y_{i,t})\right). \tag{29}$$

Using the inequality $e^{-x} \leq 1 - x + \frac{1}{2}x^2, \forall x \geq 0$, from (29) we arrive at

$$\frac{\alpha_{i,t+1} + \beta_{i,t+1}}{\alpha_{i,t} + \beta_{i,t}} \leq \frac{\alpha_{i,t}}{\alpha_{i,t} + \beta_{i,t}} \left(1 - \eta_c \mathcal{L}(f(\boldsymbol{x}_{i,t}; \boldsymbol{\theta}_t), y_{i,t}) + \frac{\eta_c^2}{2}\mathcal{L}^2(f(\boldsymbol{x}_{i,t}; \boldsymbol{\theta}_t), y_{i,t})\right)$$
$$+ \frac{\beta_{i,t}}{\alpha_{i,t} + \beta_{i,t}} \left(1 - \eta_c \mathcal{L}(f(\boldsymbol{x}_{i,t}; \boldsymbol{\phi}_{i,t}), y_{i,t}) + \frac{\eta_c^2}{2}\mathcal{L}^2(f(\boldsymbol{x}_{i,t}; \boldsymbol{\phi}_{i,t}), y_{i,t})\right). \tag{30}$$

Taking the logarithm from both sides of (30) and using the inequality $1 + x \leq e^x$, we get

$$\ln(\frac{\alpha_{i,t+1} + \beta_{i,t+1}}{\alpha_{i,t} + \beta_{i,t}}) \leq \frac{\alpha_{i,t}}{\alpha_{i,t} + \beta_{i,t}} \left(-\eta_c \mathcal{L}(f(\boldsymbol{x}_{i,t}; \boldsymbol{\theta}_t), y_{i,t}) + \frac{\eta_c^2}{2}\mathcal{L}^2(f(\boldsymbol{x}_{i,t}; \boldsymbol{\theta}_t), y_{i,t})\right)$$
$$+ \frac{\beta_{i,t}}{\alpha_{i,t} + \beta_{i,t}} \left(-\eta_c \mathcal{L}(f(\boldsymbol{x}_{i,t}; \boldsymbol{\phi}_{i,t}), y_{i,t}) + \frac{\eta_c^2}{2}\mathcal{L}^2(f(\boldsymbol{x}_{i,t}; \boldsymbol{\phi}_{i,t}), y_{i,t})\right). \tag{31}$$

Considering the assumption that $0 \leq \mathcal{L}(f(\boldsymbol{x}; \boldsymbol{\theta}), y) \leq 1, \forall \boldsymbol{x}, \boldsymbol{\theta}$, (31) can be relaxed to

$$\ln(\frac{\alpha_{i,t+1} + \beta_{i,t+1}}{\alpha_{i,t} + \beta_{i,t}}) \leq \frac{\alpha_{i,t}}{\alpha_{i,t} + \beta_{i,t}} \left(-\eta_c \mathcal{L}(f(\boldsymbol{x}_{i,t}; \boldsymbol{\theta}_t), y_{i,t})\right)$$
$$+ \frac{\beta_{i,t}}{\alpha_{i,t} + \beta_{i,t}} \left(-\eta_c \mathcal{L}(f(\boldsymbol{x}_{i,t}; \boldsymbol{\phi}_{i,t}), y_{i,t})\right) + \frac{\eta_c^2}{2}. \tag{32}$$

Summing (32) over time, we obtain

$$\ln(\frac{\alpha_{i,T+1} + \beta_{i,T+1}}{\alpha_{i,1} + \beta_{i,1}}) \leq \frac{\alpha_{i,t}}{\alpha_{i,t} + \beta_{i,t}} \left(-\eta_c \sum_{t=1}^{T} \mathcal{L}(f(\boldsymbol{x}_{i,t}; \boldsymbol{\theta}_t), y_{i,t})\right)$$
$$+ \frac{\beta_{i,t}}{\alpha_{i,t} + \beta_{i,t}} \left(-\eta_c \sum_{t=1}^{T} \mathcal{L}(f(\boldsymbol{x}_{i,t}; \boldsymbol{\phi}_{i,t}), y_{i,t})\right) + \frac{\eta_c^2 T}{2}. \tag{33}$$

According to Hölder's inequality, for any positive real numbers $p$ and $q$ satisfying $\frac{1}{p} + \frac{1}{q} = 1$, the following inequality holds:

$$\frac{\alpha_{i,T+1}}{p} + \frac{\beta_{i,T+1}}{q} \geq \alpha_{i,T+1}^{\frac{1}{p}} \beta_{i,T+1}^{\frac{1}{q}}. \tag{34}$$

To meet the condition $\frac{1}{p} + \frac{1}{q} = 1$, it is necessary that $p \geq 1$ and $q \geq 1$. Consequently, (34) can be modified to:

$$\alpha_{i,T+1} + \beta_{i,T+1} \geq \alpha_{i,T+1}^{\frac{1}{p}} \beta_{i,T+1}^{\frac{1}{q}}. \tag{35}$$

Considering the fact that $\alpha_{i,1} = \beta_{i,1} = 1$, based on (35) we can write

$$\ln(\frac{\alpha_{i,T+1} + \beta_{i,T+1}}{\alpha_{i,1} + \beta_{i,1}}) \geq \frac{1}{p}\ln(\alpha_{i,T+1}) + \frac{1}{q}\ln(\beta_{i,T+1}) - \ln(2). \tag{36}$$

According to the update rule in (29), (36) is equivalent to

$$\ln(\frac{\alpha_{i,T+1} + \beta_{i,T+1}}{\alpha_{i,1} + \beta_{i,1}}) \geq -\frac{\eta_c}{p}\sum_{t=1}^{T}\mathcal{L}(f(\boldsymbol{x}_{i,t}; \boldsymbol{\theta}_t), y_{i,t}) - \frac{\eta_c}{q}\sum_{t=1}^{T}\mathcal{L}(f(\boldsymbol{x}_{i,t}; \boldsymbol{\phi}_{i,t}), y_{i,t}) - \ln(2). \tag{37}$$

Combining (33) with (37) we arrive at

$$\frac{\alpha_{i,t}}{\alpha_{i,t}+\beta_{i,t}}\sum_{t=1}^{T}\mathcal{L}(f(\boldsymbol{x}_{i,t};\boldsymbol{\theta}_t),y_{i,t})+\frac{\beta_{i,t}}{\alpha_{i,t}+\beta_{i,t}}\sum_{t=1}^{T}\mathcal{L}(f(\boldsymbol{x}_{i,t};\boldsymbol{\phi}_{i,t}),y_{i,t})$$

$$-\frac{1}{p}\sum_{t=1}^{T}\mathcal{L}(f(\boldsymbol{x}_{i,t};\boldsymbol{\theta}_t),y_{i,t})-\frac{1}{q}\sum_{t=1}^{T}\mathcal{L}(f(\boldsymbol{x}_{i,t};\boldsymbol{\phi}_{i,t}),y_{i,t})\leq\frac{\ln(2)}{\eta_c}+\frac{\eta_c T}{2}. \tag{38}$$

Due to the convexity of $\mathcal{L}(\cdot,\cdot)$, we can write

$$\mathcal{L}(f_{i,t}(\boldsymbol{x}_{i,t}),y_{i,t})=\mathcal{L}\left(\frac{\alpha_{i,t}}{\alpha_{i,t}+\beta_{i,t}}f(\boldsymbol{x}_{i,t};\boldsymbol{\theta}_t)+\frac{\beta_{i,t}}{\alpha_{i,t}+\beta_{i,t}}f(\boldsymbol{x}_{i,t};\boldsymbol{\phi}_{i,t}),y_{i,t}\right)$$

$$\leq\frac{\alpha_{i,t}}{\alpha_{i,t}+\beta_{i,t}}\sum_{t=1}^{T}\mathcal{L}(f(\boldsymbol{x}_{i,t};\boldsymbol{\theta}_t),y_{i,t})+\frac{\beta_{i,t}}{\alpha_{i,t}+\beta_{i,t}}\sum_{t=1}^{T}\mathcal{L}(f(\boldsymbol{x}_{i,t};\boldsymbol{\phi}_{i,t}),y_{i,t}). \tag{39}$$

Combining (38) with (39) we get

$$\sum_{t=1}^{T}\mathcal{L}(f_{i,t}(\boldsymbol{x}_{i,t}),y_{i,t})-\frac{1}{p}\sum_{t=1}^{T}\mathcal{L}(f(\boldsymbol{x}_{i,t};\boldsymbol{\theta}_t),y_{i,t})-\frac{1}{q}\sum_{t=1}^{T}\mathcal{L}(f(\boldsymbol{x}_{i,t};\boldsymbol{\phi}_{i,t}),y_{i,t})$$

$$\leq\frac{\ln(2)}{\eta_c}+\frac{\eta_c T}{2} \tag{40}$$

Substituting $p=\infty$ and $q=1$ in (40), we obtain

$$\sum_{t=1}^{T}\mathcal{L}(f_{i,t}(\boldsymbol{x}_{i,t}),y_{i,t})-\sum_{t=1}^{T}\mathcal{L}(f(\boldsymbol{x}_{i,t};\boldsymbol{\phi}_{i,t}),y_{i,t})\leq\frac{\ln(2)}{\eta_c}+\frac{\eta_c T}{2}. \tag{41}$$

Since Fed-POE updates $\phi_{i,t}$ locally using online gradient descent as outlined in (8), according to (6), for any $\phi$ we can write

$$\sum_{t=1}^{T}\mathcal{L}(f(\boldsymbol{x}_{i,t};\boldsymbol{\phi}_{i,t}),y_{i,t})-\sum_{t=1}^{T}\mathcal{L}(f(\boldsymbol{x}_{i,t};\boldsymbol{\phi}),y_{i,t})\leq\frac{\|\boldsymbol{\phi}\|^2}{2\eta}+\frac{\eta}{2}G^2 T. \tag{42}$$

Substituting $\phi_i^*$ in (42) along with combining (41) with (42), we can conclude that

$$\sum_{t=1}^{T}\mathcal{L}(f_{i,t}(\boldsymbol{x}_{i,t}),y_{i,t})-\sum_{t=1}^{T}\mathcal{L}(f(\boldsymbol{x}_{i,t};\boldsymbol{\phi}_i^*),y_{i,t})\leq\frac{\|\boldsymbol{\phi}_i^*\|^2}{2\eta}+\frac{\ln(2)}{\eta_c}+\frac{\eta}{2}G^2 T+\frac{\eta_c T}{2} \tag{43}$$

which proves the personalized regret upper bound of Fed-POE in (12). Furthermore, plunging in $p=1$ and $q=\infty$ into (40), we get

$$\sum_{t=1}^{T}\mathcal{L}(f_{i,t}(\boldsymbol{x}_{i,t}),y_{i,t})-\sum_{t=1}^{T}\mathcal{L}(f(\boldsymbol{x}_{i,t};\boldsymbol{\theta}_t),y_{i,t})\leq\frac{\ln(2)}{\eta_c}+\frac{\eta_c T}{2}. \tag{44}$$

Summing (44) over all clients, we obtain

$$\sum_{t=1}^{T}\sum_{i=1}^{N}\mathcal{L}(f_{i,t}(\boldsymbol{x}_{i,t}),y_{i,t})-\sum_{t=1}^{T}\sum_{i=1}^{N}\mathcal{L}(f(\boldsymbol{x}_{i,t};\boldsymbol{\theta}_t),y_{i,t})\leq\frac{N\ln(2)}{\eta_c}+\frac{\eta_c NT}{2}. \tag{45}$$

Combining (45) with (28), we arrive at

$$\sum_{t=1}^{T}\sum_{i=1}^{N}\mathcal{L}(f_{i,t}(\boldsymbol{x}_{i,t}),y_{i,t})-\sum_{t=1}^{T}\sum_{i=1}^{N}\mathcal{L}(f(\boldsymbol{x}_{i,t};\boldsymbol{\theta}^*),y_{i,t})$$

$$\leq\frac{N\|\boldsymbol{\theta}^*\|^2}{2\eta}+\frac{N\ln(2)}{\eta_c}+\frac{\eta N}{2}G^2 T+\frac{\eta_c NT}{2} \tag{46}$$

which proves the Theorem.

# C  Proof of Theorem 3

According to assumption A3 that $0 \leq \mathcal{L}(f(\boldsymbol{x};\boldsymbol{\theta}),y) \leq 1$, it can be written that

$$\frac{1}{N}\sum_{t=1}^{U}\sum_{i=1}^{N}\mathcal{L}(f_{i,t}(\boldsymbol{x}_{i,t}),y_{i,t}) - \frac{1}{N}\sum_{t=1}^{U}\sum_{i=1}^{N}\mathcal{L}(h^{*}(\boldsymbol{x}_{i,t}),y_{i,t}) \leq U. \tag{47}$$

Let $\ell_{ij,t}$ denote the importance sampling loss estimate, which is expressed as

$$\ell_{ij,t} = \frac{\mathcal{L}(f(\boldsymbol{x}_{i,t};\boldsymbol{\rho}_{j}),y_{i,t})}{q_{ij,t}}\mathbf{1}_{j\in\mathbb{M}_{i,t}}. \tag{48}$$

Let the total number of model parameters stored by the server after time step $U$ is $D$ and $W_{i,t} = \sum_{j=1}^{D}w_{ij,t}$. For any $t > U$, considering (17), we can write

$$\frac{W_{i,t+1}}{W_{i,t}} = \sum_{j=1}^{D}\frac{w_{ij,t}}{W_{i,t}}\exp(-\eta_{c}\ell_{ij,t}) = \sum_{j=1}^{D}p_{ij,t}\exp(-\eta_{c}\ell_{ij,t}). \tag{49}$$

Employing the inequality $e^{-x} \leq 1 - x + \frac{1}{2}x^{2}, \forall x \geq 0$, from (49) we obtain

$$\frac{W_{i,t+1}}{W_{i,t}} \leq \sum_{j=1}^{D}p_{ij,t}(1 - \eta_{c}\ell_{ij,t} + \frac{\eta_{c}^{2}}{2}\ell_{ij,t}^{2}). \tag{50}$$

Taking the logarithm from both sides of (50) and using the inequality $1 + x \leq e^{x}$, we arrive at

$$\ln\frac{W_{i,t+1}}{W_{i,t}} \leq \sum_{j=1}^{D}p_{ij,t}(-\eta_{c}\ell_{ij,t} + \frac{\eta_{c}^{2}}{2}\ell_{ij,t}^{2}). \tag{51}$$

Summing (51) over time, we get

$$\ln\frac{W_{i,T+1}}{W_{i,U}} \leq \sum_{t=U}^{T}\sum_{j=1}^{D}p_{ij,t}(-\eta_{c}\ell_{ij,t} + \frac{\eta_{c}^{2}}{2}\ell_{ij,t}^{2}). \tag{52}$$

Considering the fact that the weights $\{w_{ij,t}\}_{j=1}^{D}$ are initialized as $w_{ij,1} = 1, \forall j \in [D]$, it can concluded that $W_{i,U} \leq D$. Theefore, for any $j \in [D]$, the left hand side of (52) is bounded from below as

$$\ln\frac{W_{i,T+1}}{W_{i,U}} \geq \ln\frac{w_{ij,T+1}}{W_{i,U}} \geq \ln\frac{w_{ij,T+1}}{D} = -\sum_{t=U}^{T}\eta_{c}\ell_{ij,t} - \ln D. \tag{53}$$

Combining (53) with (52) yields

$$\sum_{t=U}^{T}\sum_{j=1}^{D}p_{ij,t}\ell_{ij,t} - \sum_{t=U}^{T}\ell_{ij,t} \leq \frac{\ln D}{\eta_{c}} + \frac{\eta_{c}}{2}\sum_{t=U}^{T}\sum_{j=1}^{D}p_{ij,t}\ell_{ij,t}^{2}. \tag{54}$$

The expected values of $\ell_{ij,t}$ and $\ell_{ij,t}^{2}$ given observed losses up until time step $t$ can be obtained as

$$\mathbb{E}_{t}[\ell_{ij,t}] = \frac{\mathcal{L}(f(\boldsymbol{x}_{i,t};\boldsymbol{\rho}_{j}),y_{i,t})}{q_{ij,t}}\mathbb{E}_{t}[\mathbf{1}_{j\in\mathbb{M}_{i,t}}] = \mathcal{L}(f(\boldsymbol{x}_{i,t};\boldsymbol{\rho}_{j}),y_{i,t}) \tag{55a}$$

$$\mathbb{E}_{t}[\ell_{ij,t}^{2}] = \frac{\mathcal{L}(f(\boldsymbol{x}_{i,t};\boldsymbol{\rho}_{j}),y_{i,t})^{2}}{q_{ij,t}^{2}}\mathbb{E}_{t}[\mathbf{1}_{j\in\mathbb{M}_{i,t}}] = \frac{\mathcal{L}(f(\boldsymbol{x}_{i,t};\boldsymbol{\rho}_{j}),y_{i,t})^{2}}{q_{ij,t}} \leq \frac{1}{q_{ij,t}}. \tag{55b}$$

Taking the expectation from (54), we arrive at

$$\sum_{t=U}^{T}\sum_{j=1}^{D}p_{ij,t}\mathcal{L}(f(\boldsymbol{x}_{i,t};\boldsymbol{\rho}_{j}),y_{i,t}) - \sum_{t=U}^{T}\mathcal{L}(f(\boldsymbol{x}_{i,t};\boldsymbol{\rho}_{j}),y_{i,t}) \leq \frac{\ln D}{\eta_{c}} + \frac{\eta_{c}}{2}\sum_{t=U}^{T}\sum_{j=1}^{D}\frac{p_{ij,t}}{q_{ij,t}}. \tag{56}$$

Since $q_{ij,t} = 1 - (1 - p_{ij,t})^M = p_{ij,t}(1 + (1 - p_{ij,t}) + (1 - p_{ij,t})^2 + \ldots + (1 - p_{ij,t})^{M-1})$, it can be concluded that $q_{ij,t} \geq p_{ij,t}$. Therefore, (56) can be relaxed to

$$\sum_{t=U}^{T} \sum_{j=1}^{D} p_{ij,t} \mathcal{L}(f(\boldsymbol{x}_{i,t}; \boldsymbol{\rho}_j), y_{i,t}) - \sum_{t=U}^{T} \mathcal{L}(f(\boldsymbol{x}_{i,t}; \boldsymbol{\rho}_j), y_{i,t}) \leq \frac{\ln D}{\eta_c} + \frac{\eta_c}{2} D(T - U). \qquad (57)$$

According to model selection procedure adopted by Fed-POE presented in Algorithm 1, client $i$ chooses a subset of models by sampling them in $M$ rounds with replacement. Let $a_{ij,t} \geq 0$ denote the number of times that the model $j$ in $\mathbb{D}_t$ is chosen by client $i$ at time step $t$. The number of different situations for selected subset of models $\mathbb{M}_{i,t}$ is equal to the number of solutions for the linear equation

$$a_{i1,t} + \ldots + a_{iD,t} = M, a_{ij,t} \geq 0, \forall j \in [D]. \qquad (58)$$

Let $\mathbb{A}$ denote the set of all possible solutions for (58) such that if $\boldsymbol{a} \in \mathbb{A}$ where $\boldsymbol{a} = [a_1, \ldots, a_D]$, $a_1, \ldots, a_D$ satisfies (58). Therefore, for the expected loss of the ensemble $\tilde{f}_{i,t}(\boldsymbol{x}_{i,t})$ in (14), we can write

$$\mathbb{E}_t[\mathcal{L}(\tilde{f}_{i,t}(\boldsymbol{x}_{i,t}), y_{i,t})] = \sum_{k=1}^{|\mathbb{A}|} \prod_{j=1}^{D} p_{ij,t}^{a_{j,k}} \mathcal{L}(\tilde{f}_{i,t}^{(k)}(\boldsymbol{x}_{i,t}), y_{i,t}) \qquad (59)$$

where $\tilde{f}_{i,t}^{(k)}(\boldsymbol{x}_{i,t})$ denote the $k$-th possible ensemble model generated by client $i$ using Fed-POE. Using the Jensen inequality and convexity of the loss function, we can relax (59) to

$$\mathbb{E}_t[\mathcal{L}(\tilde{f}_{i,t}(\boldsymbol{x}_{i,t}), y_{i,t})] = \sum_{k=1}^{|\mathbb{A}|} \left( \prod_{j=1}^{D} p_{ij,t}^{a_{j,k}} \right) \mathcal{L}(\tilde{f}_{i,t}^{(k)}(\boldsymbol{x}_{i,t}), y_{i,t})$$

$$\leq \sum_{k=1}^{|\mathbb{A}|} \left( \prod_{j=1}^{D} p_{ij,t}^{a_{j,k}} \right) \sum_{m \in \mathbb{M}_{i,t}^{(k)}} \frac{w_{im,t}}{W_{i,t}^{(k)}} \mathcal{L}(f(\boldsymbol{x}_{i,t}; \boldsymbol{\rho}_j), y_{i,t}) \qquad (60)$$

where $\mathbb{M}_{i,t}^{(k)}$ and $W_{i,t}^{(k)}$ are the $k$-th possible model subset and weight summations, respectively. Rearranging the right hand side of (60), we can write

$$\mathbb{E}_t[\mathcal{L}(\tilde{f}_{i,t}(\boldsymbol{x}_{i,t}), y_{i,t})] \leq \sum_{j=1}^{D} p_{ij,t} \sum_{k=1}^{|\mathbb{B}_j|} \left( \prod_{m=1}^{D} p_{im,t}^{b_{m,k}} \right) \frac{w_{ij,t}}{W_{i,t}^{(k)}} \mathcal{L}(f(\boldsymbol{x}_{i,t}; \boldsymbol{\rho}_j), y_{i,t}) \qquad (61)$$

where $\mathbb{B}_j$ is the set of all possible solutions for the linear equation in (58) condition on $a_{ij,t} \geq 1$. Since for any $k$, we have $w_{ij,t} \leq W_{i,t}^{(k)}$, (61) can be relaxed to

$$\mathbb{E}_t[\mathcal{L}(\tilde{f}_{i,t}(\boldsymbol{x}_{i,t}), y_{i,t})] \leq \sum_{j=1}^{D} p_{ij,t} \sum_{k=1}^{|\mathbb{B}_j|} \left( \prod_{m=1}^{D} p_{im,t}^{b_{m,k}} \right) \mathcal{L}(f(\boldsymbol{x}_{i,t}; \boldsymbol{\rho}_j), y_{i,t}). \qquad (62)$$

Since $\sum_{k=1}^{|\mathbb{B}_j|} \left( \prod_{m=1}^{D} p_{im,t}^{b_{m,k}} \right)$ includes all possibilities in $\mathbb{B}_j$, we can conclude that $\sum_{k=1}^{|\mathbb{B}_j|} \left( \prod_{m=1}^{D} p_{im,t}^{b_{m,k}} \right) = 1$. Combining this with (62), we obtain

$$\mathbb{E}_t[\mathcal{L}(\tilde{f}_{i,t}(\boldsymbol{x}_{i,t}), y_{i,t})] \leq \sum_{j=1}^{D} p_{ij,t} \mathcal{L}(f(\boldsymbol{x}_{i,t}; \boldsymbol{\rho}_j), y_{i,t}). \qquad (63)$$

Combining (63) with (57), we arrive at

$$\sum_{t=U}^{T} \mathbb{E}_t[\mathcal{L}(\tilde{f}_{i,t}(\boldsymbol{x}_{i,t}), y_{i,t})] - \sum_{t=U}^{T} \mathcal{L}(f(\boldsymbol{x}_{i,t}; \boldsymbol{\rho}_j), y_{i,t}) \leq \frac{\ln D}{\eta_c} + \frac{\eta_c}{2} D(T - U). \qquad (64)$$

Since $0 \leq \mathcal{L}(f(\boldsymbol{x}; \boldsymbol{\theta}), y) \leq 1$, it can be written that

$$\sum_{t=1}^{U} \mathbb{E}_t[\mathcal{L}(\tilde{f}_{i,t}(\boldsymbol{x}_{i,t}), y_{i,t})] - \sum_{t=1}^{U} \mathcal{L}(f(\boldsymbol{x}_{i,t}; \boldsymbol{\rho}_j), y_{i,t}) \leq U. \qquad (65)$$

Combining (65) with (64), we get

$$\sum_{t=1}^{T} \mathbb{E}_t[\mathcal{L}(\tilde{f}_{i,t}(\boldsymbol{x}_{i,t}), y_{i,t})] - \sum_{t=1}^{T} \mathcal{L}(f(\boldsymbol{x}_{i,t}; \boldsymbol{\rho}_j), y_{i,t}) \leq \frac{\ln D}{\eta_c} + \frac{\eta_c}{2} D(T - U) + U. \quad (66)$$

Since $\bar{f}_{i,t}(\cdot)$ similar to $f_{i,t}(\cdot)$ is the ensemble of two models, following the same derivation steps from (29) to (41) by substituting $\bar{f}_{i,t}(\boldsymbol{x}_{i,t})$, $f_{i,t}(\boldsymbol{x}_{i,t})$ and $\tilde{f}_{i,t}(\boldsymbol{x}_{i,t})$ with $f_{i,t}(\boldsymbol{x}_{i,t})$, $f(\boldsymbol{x}_{i,t}; \boldsymbol{\theta}_t)$ and $f(\boldsymbol{x}_{i,t}; \boldsymbol{\phi}_{i,t})$, respectively, we can conclude that

$$\sum_{t=1}^{T} \mathcal{L}(\bar{f}_{i,t}(\boldsymbol{x}_{i,t}), y_{i,t}) - \sum_{t=1}^{T} \mathcal{L}(f_{i,t}(\boldsymbol{x}_{i,t}), y_{i,t}) \leq \frac{\ln(2)}{\eta_c} + \frac{\eta_c T}{2}, \quad (67a)$$

$$\sum_{t=1}^{T} \mathcal{L}(\bar{f}_{i,t}(\boldsymbol{x}_{i,t}), y_{i,t}) - \sum_{t=1}^{T} \mathcal{L}(\tilde{f}_{i,t}(\boldsymbol{x}_{i,t}), y_{i,t}) \leq \frac{\ln(2)}{\eta_c} + \frac{\eta_c T}{2}. \quad (67b)$$

Taking the expectation from both sides of (67b) with respect to randomization in model selection along with combining (67b) with (66) yields

$$\sum_{t=1}^{T} \mathbb{E}_t[\mathcal{L}(\bar{f}_{i,t}(\boldsymbol{x}_{i,t}), y_{i,t})] - \sum_{t=1}^{T} \mathcal{L}(f(\boldsymbol{x}_{i,t}; \boldsymbol{\rho}_j), y_{i,t})$$
$$\leq \frac{\ln 2D}{\eta_c} + \frac{\eta_c}{2}(D + 1)T + (1 - \frac{\eta_c}{2}D)U. \quad (68)$$

Furthermore, combining (67a) with (41) and (44) and taking the expectation with respect to model selection randomization, we obtain

$$\sum_{t=1}^{T} \mathbb{E}_t[\mathcal{L}(\bar{f}_{i,t}(\boldsymbol{x}_{i,t}), y_{i,t})] - \sum_{t=1}^{T} \mathcal{L}(f(\boldsymbol{x}_{i,t}; \boldsymbol{\phi}_{i,t}), y_{i,t}) \leq \frac{\ln(4)}{\eta_c} + \eta_c T, \quad (69a)$$

$$\sum_{t=1}^{T} \mathbb{E}_t[\mathcal{L}(\bar{f}_{i,t}(\boldsymbol{x}_{i,t}), y_{i,t})] - \sum_{t=1}^{T} \mathcal{L}(f(\boldsymbol{x}_{i,t}; \boldsymbol{\theta}_t), y_{i,t}) \leq \frac{\ln(4)}{\eta_c} + \eta_c T. \quad (69b)$$

Recall that $h_j(\cdot)$ associated with the $j$-th model parameter in $\mathbb{D}_t$ be defined as $h_j(\boldsymbol{x}_{i,t}) = f(\boldsymbol{x}_{i,t}; \boldsymbol{\rho}_j)$ while $h_{\text{loc}}(\cdot)$ and $h_{\text{fed}}(\cdot)$ correspond to the local and federated models, respectively, defined as $h_{\text{loc}}(\boldsymbol{x}_{i,t}) = f(\boldsymbol{x}_{i,t}; \boldsymbol{\phi}_{i,t})$ and $h_{\text{fed}}(\boldsymbol{x}_{i,t}) = f(\boldsymbol{x}_{i,t}; \boldsymbol{\theta}_t)$. Also recall that $\mathbb{H} := \{h_j \mid \forall j : 1 \leq j \leq |\mathbb{D}_T|\} \cup \{h_{\text{loc}}, h_{\text{fed}}\}$. Comparing the right hand sides of (69a) and (69b) with that of (68) and considering the fact that $D \geq 2$, for any $h \in \mathbb{H}$ we can write

$$\sum_{t=1}^{T} \mathbb{E}_t[\mathcal{L}(\bar{f}_{i,t}(\boldsymbol{x}_{i,t}), y_{i,t})] - \sum_{t=1}^{T} \mathcal{L}(h(\boldsymbol{x}_{i,t}), y_{i,t}) \leq \frac{\ln 2D}{\eta_c} + \frac{\eta_c}{2}(D + 1)T + (1 - \frac{\eta_c}{2}D)U. \quad (70)$$

By substituting $h(\cdot)$ with $h_i^*(\cdot)$ as defined in (18b) and considering the fact that $D \leq U/n$, we obtain the personalized regret upper bound for client $i$ as shown in (20). Moreover, taking the average of (70) across clients and substituting $h(\cdot)$ with $h^*(\cdot)$, we arrive at

$$\frac{1}{N} \sum_{t=1}^{T} \sum_{i=1}^{N} \mathbb{E}_t[\mathcal{L}(\bar{f}_{i,t}(\boldsymbol{x}_{i,t}), y_{i,t})] - \frac{1}{N} \sum_{t=1}^{T} \sum_{i=1}^{N} \mathcal{L}(h^*(\boldsymbol{x}_{i,t}), y_{i,t})$$
$$\leq \frac{\ln 2D}{\eta_c} + \frac{\eta_c}{2}(D + 1)T + (1 - \frac{\eta_c}{2}D)U \quad (71)$$

which proves the theorem.

## D  Supplementary Experimental Details

This section presents supplementary experimental results and details about experimental setup. All experiments were carried out using Intel(R) Core(TM) i7-10510U CPU @ 1.80 GHz 2.30 GHz processor with a 64-bit Windows operating system.

### D.1 Regression Data Distribution

As it is pointed out in section 5, the present paper tests the performance of algorithms for online regression task on Air and WEC datasets. These datasets are downloaded from UCI Machine Learning Repository [27]. Each data sample in Air dataset includes air quality information such as concentration of some chemicals in the air. Data samples in Air dataset collected from four different geographical locations. Moreover, data samples in WEC, collected from wave energy converters in four different geographical locations. In order to distribute data, clients are partitioned into 4 groups. For each group, $70\%$ of data samples observed by each client in the group belongs to a specific geographical location while $10\%$ of observed data samples belong to each of the rest 3 locations.

### D.2 Random Feature Kernel-based Models

As it is pointed in section 5, the proposed Fed-POE and all baselines utilize a random feature kernel-based model to perform online regression task. In what follows we explain random feature-based kernel models. Let $\kappa(\cdot, \cdot)$ be a positive-definite function called kernel such that $\kappa(\boldsymbol{x}, \boldsymbol{x}')$ measures the similarity between $\boldsymbol{x}$ and $\boldsymbol{x}'$. In online kernel learning context, at time step $t + 1$, the following prediction is made for $\boldsymbol{x}$ (see e.g. [52, 25, 48]):

$$f_\kappa(\boldsymbol{x}; \boldsymbol{\alpha}_t) = \sum_{\tau=1}^{t} \sum_{i=1}^{N} \alpha_{i,\tau} \kappa(\boldsymbol{x}, \boldsymbol{x}_{i,\tau}) \tag{72}$$

where $\boldsymbol{\alpha}_t = [\alpha_{1,1}, \ldots, \alpha_{N,1}, \ldots, \alpha_{1,t}, \ldots, \alpha_{N,t}]$ denotes the learnable parameters. Therefore, the number of parameters that should be learned grows with time. In order to alleviate the computational complexity of online kernel learning, random feature approximation [44] can be employed. In fact, using random feature approximation, the number of parameters that needs be learned is time-invariant and is selected by the algorithm. Assume that $\kappa(\cdot)$ is a shift-invariant kernel function such that $\kappa(\boldsymbol{x}, \boldsymbol{x}') = \kappa(\boldsymbol{x} - \boldsymbol{x}')$. Also, suppose that $\kappa(\cdot)$ is scaled such that $\kappa(\boldsymbol{0}) = 1$. Let $\xi(\cdot)$ denotes the Fourier transform of $\kappa(\cdot)$. According to definition of inverse Fourier transform $\kappa(\boldsymbol{0}) = \int_{-\infty}^{\infty} \xi(\boldsymbol{\omega}) d\boldsymbol{\omega} = 1$. Therefore, it can be concluded that $\xi(\cdot)$ is a probability density function (PDF). Let $\boldsymbol{\omega}_1, \ldots, \boldsymbol{\omega}_D$ be drawn randomly from $\xi(\cdot)$ and called random features. Using the random features $\boldsymbol{\omega}_1, \ldots, \boldsymbol{\omega}_D$, the representation $\boldsymbol{z}(\boldsymbol{x})$ is defined as

$$\boldsymbol{z}(\boldsymbol{x}) = \frac{1}{\sqrt{D}} [\sin(\boldsymbol{\omega}_1^\top \boldsymbol{x}), \ldots, \sin(\boldsymbol{\omega}_D^\top \boldsymbol{x}), \cos(\boldsymbol{\omega}_1^\top \boldsymbol{x}), \ldots, \cos(\boldsymbol{\omega}_D^\top \boldsymbol{x})]. \tag{73}$$

Given random features $\boldsymbol{\omega}_1, \ldots, \boldsymbol{\omega}_D$ and using the proposed Fed-POE, at time step $t$, client $i$ makes prediction $f(\boldsymbol{x}_{i,t}; \boldsymbol{\theta}_t) = \boldsymbol{\theta}_t^\top \boldsymbol{z}(\boldsymbol{x}_{i,t})$. Clients and the server employ the proposed Fed-POE to learn the parameter $\boldsymbol{\theta}_t$. As it can be inferred since the model $f(\cdot; \cdot)$ is linear with respect to model parameter $\boldsymbol{\theta}_t$, using convex loss functions, the loss $\mathcal{L}(f(\boldsymbol{x}_{i,t}; \boldsymbol{\theta}_t), y_{i,t})$ is convex as well.

Furthermore, in section 5, the proposed Fed-POE utilizes three Gaussian kernels for online regression on Air and WEC datasets. In order to implement multi-kernel learning for Fed-POE, the prediction of kernels are linearly combined and the weights for linear combination is learned locally by each client. Let $f_{0.1}(\boldsymbol{x}_{i,t})$, $f_1(\boldsymbol{x}_{i,t})$ and $f_{10}(\boldsymbol{x}_{i,t})$ represent predictions of Gaussian kernels with variances of 0.1, 1 and 10, respectively. Then at time step $t$, client $i$ makes prediction $w_{0.1,it} f_{0.1}(\boldsymbol{x}_{i,t}) + w_{1,it} f_1(\boldsymbol{x}_{i,t}) + w_{10,it} f_{10}(\boldsymbol{x}_{i,t})$. In order to update weights $w_{0.1,it}$, $w_{1,it}$ and $w_{10,it}$, client $i$ employs multiplicative update rule. As an example after observing the loss $\mathcal{L}(f_1(\boldsymbol{x}_{i,t}), y_{i,t})$, client $i$ updates $w_{1,it}$ as $w_{1,i(t+1)} = w_{1,it} \exp(-\gamma_i \mathcal{L}(f_1(\boldsymbol{x}_{i,t}), y_{i,t}))$ where $\gamma_i$ is a learning rate specified by client $i$.

### D.3 Image Classification Experimental Setup

The pre-trained CNN used by Fed-POE and other baselines is biased toward class label 0. For CIFAR-10, the pre-trained CNN is trained on a subset of the CIFAR-10 training data, consisting of $5,000$ samples with label 0 and 500 samples from each of the other 9 class labels. For FMNIST, the model is trained on a subset of the FMNIST training data, consisting of $6,000$ samples with label 0 and 500 samples from each of the other 9 class labels. The CNN models are trained using Tensorflow 2.16.1. We used the SGD optimizer with a learning rate of $10^{-3}$ and momentum of 0.9. The models for CIFAR-10 and FMNIST were trained for 100 epochs and 10 epochs, respectively.

Clients receive test data samples sequentially and make prediction for the newly received sample. To distribute test data samples of CIFAR-10 among clients, we split clients into 10 groups. For

Table 4: Average accuracy and standard deviation across clients employing Fed-POE for image classification with varying values of $M$

| **Datasets** | $M = 0$ | $M = 4$ | $M = 8$ | $M = 16$ |
|---|---|---|---|---|
| CIFAR-10 | $65.55\% \pm 8.77\%$ | $66.50\% \pm 8.00\%$ | $\mathbf{66.54\% \pm 8.08\%}$ | $66.46\% \pm 7.98\%$ |
| FMNIST | $79.03\% \pm 1.76\%$ | $79.12\% \pm 1.87\%$ | $\mathbf{79.23\% \pm 1.88\%}$ | $79.18\% \pm 1.85\%$ |

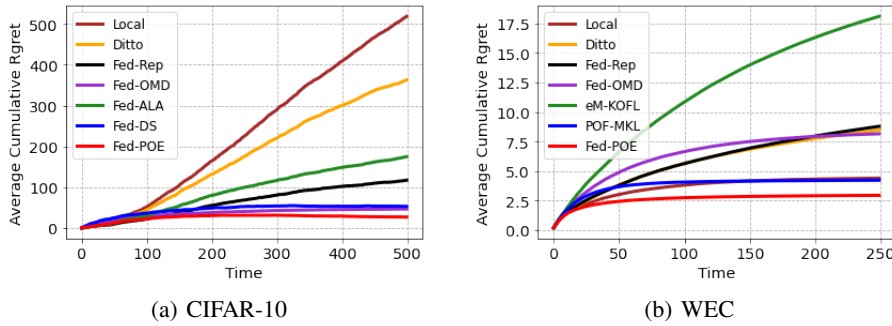

(a) CIFAR-10          (b) WEC

Figure 1: Cumulative regret over time on CIFAR-10 and WEC datasets.

CIFAR-10, $55\%$ of samples observed by a client belongs to a specific class label while only $5\%$ of received samples belong to each of other 9 class labels. For FMNIST, client data distribution is time-variant. Since the number of test samples is $10,000$ and the number of clients is $20$, it can be concluded that time horizon $T$ is 500. In the first half of time steps (i.e. $t \leq 250$), each client observes 200 samples from the first 5 class labels and 50 samples from other 5 class labels. In the second half, this is reversed: clients observe 200 samples from the last 5 class labels and 50 samples from the other class labels. In each half, each client is biased toward one of the five majority classes. For example, if a client is biased toward class 0 in the first half, it observes 100 samples from class label 0, 25 samples from each of class labels 1 to 4, and 10 samples from each of class labels 5 to 9. Therefore, in each half, each client observes 100 samples from one class, 25 samples from each of four other classes, and 10 samples from each of the remaining five classes.

To implement Fed-OMD for both regression and image classification, we used the $\ell_2$-norm as a regularizer function for mirror descent. For implementing Ditto for both regression and image classification, we set the regularization factor $\lambda$ to 1. In the case of image classification using Fed-Rep, clients locally fine-tune the last two layers of the CNN model, while the rest of the network is used as the global backbone to generate representations. Furthermore, Fed-POE is compatible with any federated learning method and can utilize any federated algorithm. For CIFAR-10, Fed-POE uses Fed-OMD, while for FMNIST, Fed-POE uses Fed-Rep. To fine-tune the CNN model, Fed-POE and all baselines use the SGD optimizer and the cross-entropy loss function.

### D.4 Supplementary Results

Table 4 presents the accuracy of clients for image classification using Fed-POE with varying values of $M$. As can be seen for both CIFAR-10 and FMNIST, when $M > 0$, the accuracy is higher than in the case where $M = 0$. The case where $M = 0$ corresponds to clients not using models stored by the server in their ensemble. Therefore, these results show that constructing the ensemble using previous federated model parameters stored by the server improves the accuracy of Fed-POE. This indicates the effectiveness of the model selection procedure of Fed-POE presented in Algorithm 1. Additionally, these results show that increasing $M$ does not necessarily lead to further accuracy improvement. This implies the effectiveness of Fed-POE's model selection in pruning model parameters from the ensemble that have relatively lower accuracy. Figure 1 illustrates the average cumulative global regret of clients over time using Fed-POE and all other baselines. As depicted, Fed-POE achieves sublinear regret for both CIFAR-10 and WEC datasets. This corroborates the theoretical results in Theorems 2 and 3.

