# OpenReview forum: "Personalized Federated Learning with Mixture of Models for Adaptive Prediction and Model Fine-Tuning"
_NeurIPS.cc/2024/Conference — NeurIPS 2024 poster_

### Official Review · Reviewer_rUKC · 2024-06-30

**Soundness:** 3
**Presentation:** 3
**Contribution:** 2
**Rating:** 4
**Confidence:** 5

**Summary:**

This paper introduces a personalized federated learning algorithm to address the challenges of real-time predictions in non-stationary environments. Clients fine-tune models online, combining their locally fine-tuned models with multiple federated models learned over time. This approach ensures efficient adaptation to evolving data streams, with theoretical analysis and experiments on real datasets demonstrating its effectiveness.

**Strengths:**

- The proposed algorithm effectively addresses the challenge of making real-time predictions in non-stationary environments by allowing clients to fine-tune models online, ensuring continuous adaptation to evolving data streams.
- By combining locally fine-tuned models with multiple federated models, the approach enhances personalization and leverages the strengths of both local and federated learning, resulting in improved performance.
- The paper provides a solid theoretical analysis alongside experimental validation on real datasets, demonstrating the practical effectiveness and robustness of the proposed algorithm in real-world scenarios.

**Weaknesses:**

1. Contributions are suggested to list by items for clear summaries.
2. The baselines in Table 1 are all before the 2022 year, more latest related methods published in 2023 should be compared.
3. Fed-POE has limited improvements on Air and FMNIST datasets.
4. The process of combining locally fine-tuned models with multiple federated models may introduce significant computational overhead for clients, especially those with limited resources.
5. As the number of clients increases, managing and integrating multiple personalized models can become complex, posing scalability challenges for the proposed algorithm.

**Questions:**

No

**Limitations:**

Yes

---

> ### Author Rebuttal · Authors · 2024-08-07
>
> Thank you very much for taking the time to review our paper and providing your valuable comments. Please find below our responses to your comments and questions.
>
> We will revise the presentation of the contributions section in the introduction by breaking down the last paragraph into itemized points.
>
> ## Baselines in Table 1
> For the results in Table 1, we used online kernel-based models to evaluate the performance of Fed-POE in a convex setting. We believe that we compared the performance of Fed-POE against all state-of-the-art online federated kernel learning algorithms which provide rigorous theoretical guarantees. However, all these works were for the year 2022 and before. Moreover, if one employs Fed-DS, which is a baseline from 2023, for the online federated kernel learning problem investigated in Table 1, it is equivalent to Fed-OMD in this specific case. Therefore, our results in Table 1 are relevant to the baselines from 2023.
>
> ## Improvements on Air and FMNIST Datasets
> We believe the performance of the proposed Fed-POE on the Air and FMNIST datasets demonstrates its advantage in real-time prediction tasks. The main challenge in real-time prediction arises when there is no prior information about the streaming data, making it difficult to evaluate model performance before the task begins. As discussed in Section 3.2 of the paper, it is theoretically unclear whether federated or local models perform better under such conditions. The performance largely depends on the dataset.
>
> Results presented in Section 5 confirm that the relative performance of algorithms varies with the dataset. Tables 1 and 2 show that for the CIFAR-10 dataset, federated models outperform local models, while for the Air dataset, local models perform better than federated models. Furthermore, for WEC and FMNIST, both local models and personalized federated learning models outperform non-personalized federated models. This variability complicates the decision between local and federated models. However, the results in Tables 1 and 2 indicate that Fed-POE consistently outperforms all baselines, albeit marginally on some datasets. This suggests that Fed-POE's performance is robust across different datasets, making it a reliable choice for real-time prediction tasks in the absence of prior information.
>
> ## Complexity
> We would like to clarify that at each time step, Fed-POE fine-tunes only one federated model while making inferences with multiple federated models. Given that the computational complexity of making inferences is usually considerably less than that of model fine-tuning, we believe Fed-POE does not introduce significant additional computational overhead. Furthermore, the number of federated models used by clients can be adjusted to ensure that the required computation and memory costs remain manageable for the clients. We analyze the computational complexity of Fed-POE.
>
> Let $C_F$ denote the number of computations required to fine-tune the model $f$, and let $C_I$ represent the number of computations required to make an inference with model $f$. Assume that the complexity of model selection in Algorithm 1 is negligible compared to fine-tuning and making inferences with model $f$. According to Algorithm 2, each client performs $2C_F + (M+2)C_I$ computations per time step. Therefore, the computational complexity of Fed-POE for each client is $\mathcal{O}(C_F + MC_I)$. Typically, fine-tuning deep neural networks with backpropagation requires significantly more computations than making inferences with them. Thus, if the model $f$ is a deep neural network, $C_I$ is negligible compared to $C_F$. In this case, the computational complexity for each client using Fed-POE is $\mathcal{O}(C_F)$, which is comparable to most state-of-the-art federated learning methods.
>
> ## Increase in the Number of Clients
> An increase in the number of clients does **not** pose scalability issues for Fed-POE compared to other state-of-the-art federated learning methods. As with most other federated learning algorithms, each client using Fed-POE only fine-tunes one model at each step and sends the update to the server.

---

### Official Review · Reviewer_extd · 2024-07-11

**Soundness:** 2
**Presentation:** 2
**Contribution:** 2
**Rating:** 5
**Confidence:** 4

**Summary:**

This paper proposes a novel personalized federated learning algorithm, Fed-POE, which is designed for adaptive prediction and model fine-tuning in dynamic environments. It addresses the challenge of real-time predictions on streaming data by constructing a personalized model that combines a locally fine-tuned model with multiple federated models. Theoretical analysis and experiments on real datasets demonstrate its effectiveness in achieving sublinear regret bounds and improved online prediction accuracy.

**Strengths:**

1. The paper proposes a unique ensemble method that dynamically combines local and federated models, which is a novel approach in the field of federated learning.

2. It provides a solid theoretical analysis, demonstrating sublinear regret bounds for convex models.

3. The paper is well organized.

**Weaknesses:**

1. Although the presented method is novel, it is simply a combination of the previous personalized federated learning approaches as well as ensemble learning and provides comparably little conceptual originality.  The contribution's main novelty seems to be that integrating results from prior models would be beneficial in mitigating catastrophic forgetting in online federated learning.

2. Experimental results show that the improvement in the accuracy of Fed-POE compared to other methods is not significant, but ensemble learning inevitably increases the computational overhead increase. The paper needs to analyze whether this trade-off is reasonable.

3. The paper needs more experiments to prove the effectiveness of the method, for example, for real-time predictions, the size of the old data replay is crucial, and the authors should design experiments to analyze the effect of batch size b on the experimental results. This paper also needs experiment results on the accuracy over the time step.

**Questions:**

1. The method in this paper does not significantly improve accuracy and even has a larger standard deviation. Can you give me more reasons to support your methods？

2. The method is designed with two parts to mitigate catastrophic forgetting (old data replay and integration of multiple old models), The complex model updating process is unreasonable for real-time prediction. Can you design more ablation experiments to analyze these two parts？

**Limitations:**

see the weaknesses and questions

---

> ### Author Rebuttal · Authors · 2024-08-07
>
> Thank you very much for taking the time to review our paper and providing your valuable comments. Please find below our responses to your comments and questions.
>
> We would like to briefly review the main contributions of this paper, which we believe are of interest and utility to the community. We hope the respected reviewer will consider these contributions.
>
> Section 3 of the paper highlights the challenge of choosing between local models and federated models in online federated learning when there is no prior information about data distribution. In Section 4.1, the paper proposes an ensemble federated learning algorithm that effectively leverages both local and federated models to provide more reliable performance for clients. Moreover, Section 4.2 demonstrates how federated learning can be employed to address catastrophic forgetting in real-time decision making.
>
> ## Performance Gain
> We believe the performance of the proposed Fed-POE in Section 5 demonstrates its advantage in real-time prediction tasks. The main challenge in real-time prediction arises when there is no prior information about the streaming data, making it difficult to evaluate model performance before the task begins. As discussed in Section 3.2 of the paper, it is theoretically unclear whether federated or local models perform better under such conditions. The performance largely depends on the dataset.
>
> Results presented in Section 5 confirm that the relative performance of algorithms varies with the dataset. Tables 1 and 2 show that for the CIFAR-10 dataset, federated models outperform local models, while for the Air dataset, local models perform better than federated models. Furthermore, for WEC and FMNIST, both local models and personalized federated learning models outperform non-personalized federated models. This variability complicates the decision between local and federated models. However, the results in Tables 1 and 2 indicate that Fed-POE consistently outperforms all baselines, albeit marginally on some datasets. This suggests that Fed-POE's performance is robust across different datasets, making it a reliable choice for real-time prediction tasks in the absence of prior information.
>
> ## Further Ablation Study
> To address your concerns, we performed additional ablation studies to analyze the effect of batch size $b$ on the Fed-POE performance. We conducted experiments on the CIFAR-10 dataset, varying the batch size $b$ and the number of models $M$ selected by each client to construct the ensemble model. The table below illustrates the results. As observed, the batch size $b=1$ results in the worst accuracy, mainly due to the forgetting process where models overfit to the most recently observed data. However, increasing the batch size from $b=10$ or $b=20$ to $b=30$ does not significantly improve the accuracy. Larger batch sizes may lead the model to perform better on older data, as the model is trained on older data over more iterations. Therefore, from this study, we conclude that a moderate batch size is optimal, considering that an increase in batch size leads to an increase in computational complexity. Based on these findings, we choose $b=10$. Furthermore, in Figure 1 in Appendix D, we illustrate the regret of Fed-POE and other baselines over time for the CIFAR-10 and WEC datasets.
>
> | CIFAR-10    | M=0 | M=4 | M=8 | M=16 |
> | -------- | ------- | -------- | -------- | -------- |
> | $b=1$  | 53.80% $\pm$ 6.71%    | 62.73%$\pm$8.29% | 62.73%$\pm$8.29% | 62.73%$\pm$8.26% |
> | $b = 10$ | 65.55%$\pm$8.77%     | 66.50%$\pm$8.00% | 66.54%$\pm$8.08% | 66.46%$\pm$7.98% |
> | $b = 20$    | 65.72%$\pm$8.62%   | 66.13%$\pm$8.20% | 66.64%$\pm$7.94% | 66.53%$\pm$8.00% |
> | $b = 30$ | 65.83%$\pm$8.54% | 66.32%$\pm$7.92% | 66.24%$\pm$8.05% | 66.39%$\pm$8.02% |
>
> ## Complexity
> Similar to other federated learning algorithms, using Fed-POE each client only updates one federated model per step. Therefore, we believe that using Fed-POE does not impose significant additional updating complexity compared to other federated learning counterparts. Please find below the computational complexity analysis of Fed-POE.
>
> Let $C_F$ denote the number of computations required to fine-tune the model $f$, and let $C_I$ represent the number of computations required to make an inference with model $f$. Assume that the complexity of model selection in Algorithm 1 is negligible compared to fine-tuning and making inferences with model $f$. According to Algorithm 2, each client performs $2C_F + (M+2)C_I$ computations per time step. Therefore, the computational complexity of Fed-POE for each client is $\mathcal{O}(C_F + MC_I)$. Typically, fine-tuning deep neural networks with backpropagation requires significantly more computations than making inferences with them. Thus, if the model $f$ is a deep neural network, $C_I$ is negligible compared to $C_F$. In this case, the computational complexity for each client using Fed-POE is $\mathcal{O}(C_F)$, which is comparable to most state-of-the-art federated learning methods.

---

> > ### Comment · Reviewer_extd · 2024-08-13
> >
> > Thanks for the authors' detailed feedback. In light of these explanations, I will revise my score accordingly.

---

> > > ### Author Response · Authors · 2024-08-13
> > >
> > > Thank you very much for your feedback. We would be happy to address any further concerns or questions you may have.

---

### Official Review · Reviewer_NAbk · 2024-07-14

**Soundness:** 3
**Presentation:** 3
**Contribution:** 3
**Rating:** 6
**Confidence:** 4

**Summary:**

The paper introduces an interesting perspective about the role of ensembles of models in federated learning. The provocative claim is the fact that federated learning is not always better than locally-trained models. This is contextualized in the field of not IID data and time-varying data generating processes. To address this issue the paper introduces from the theoretical point of view how to quantify the regret in federated and locally trained models. In addition it includes an analysis of non-convex models by managing an history of models. The overall impression about the paper is positive even if some points could have been better explored (in particular the part related to not IID data that is somehow the core of the paper).

**Strengths:**

- The paper introduces a theoretical evaluation of the gain produced by federated models w.r.t. locally trained models. This results show that federated learning is relevant only when models can be considered iid (hence averaging providing better results). This is somehow a know results but I appreciated the theoretical analysis
- The proposed solution is to combine with a convex mean a locally-trained models with the federated models
- This is further extended in case of non-convex models by considering an "history" of models to be used when needed (i.e., according to the loss)

**Weaknesses:**

- The federated models somehow includes the locally-trained model. I would have appreciated a further analysis about the fact that the two "sides" of the average model are related each other.
- The setting in which eta and eta_c scales with T prevents adaptation in the long run (which is somehow the core of the paper). How to deal with that?
- Federated learning typically takes also into account the complexity of the learning phase (i.e., the amount of info to be transmitted, e.g., the models). This is not quantified here. And this could be also a weak point in the fed-poe algorithm.

**Questions:**

See Weaknesses box

**Limitations:**

See Weaknesses box

---

> ### Author Rebuttal · Authors · 2024-08-07
>
> Thank you very much for taking the time to review our paper and providing your valuable comments. Please find below our responses to your comments and questions.
>
> ## Relations between Federated Models and Local Models
> The federated model can differ significantly from the local models, especially when the data distribution among clients is heterogeneous. The degree of data heterogeneity influences the similarity between the federated model and the local models. Although the federated model is trained on local models, if the gradient trajectories of other clients differ significantly from those of a particular client, it is expected that the federated model will be substantially different from the local model of that outlier client. We can add this to the paper.
>
> ## Learning Rates in the Long Run
> If the time horizon $T$ is unknown which may often be the case in the long run, the doubling trick technique (see e.g., [R1]) can be effectively used to set the learning rates $\eta$ and $\eta_c$ while maintaining theoretical guarantees. The doubling trick is a well-known technique in online learning that adaptively sets the learning rates without knowing the time horizon. We add a note about this to the paper.
>
> [R1] N. Alon, N. Cesa-Bianchi, C. Gentile, S. Mannor, Y. Mansour, and O. Shamir, “Nonstochastic multi-armed bandits with graph-structured feedback,” SIAM J. Comput., vol. 46, no. 6, pp. 1785–1826, 2017.
>
> ## Complexity
> Let $C_F$ denote the number of computations required to fine-tune the model $f$, and let $C_I$ represent the number of computations required to make an inference with model $f$. Assume that the complexity of model selection in Algorithm 1 is negligible compared to fine-tuning and making inferences with model $f$. According to Algorithm 2, each client performs $2C_F + (M+2)C_I$ computations per time step. Therefore, the computational complexity of Fed-POE for each client is $\mathcal{O}(C_F + MC_I)$. Typically, fine-tuning deep neural networks with backpropagation requires significantly more computations than making inferences with them. Thus, if the model $f$ is a deep neural network, $C_I$ is negligible compared to $C_F$. In this case, the computational complexity for each client using Fed-POE is $\mathcal{O}(C_F)$, which is comparable to most state-of-the-art federated learning methods.
>
> Moreover, at each time step, each client sends one updated model to the server. Let $P_f$ denote the number of parameters in model $f$. Therefore, the amount of information that needs to be transmitted to the server is $\mathcal{O}(P_f)$, which is the same as most state-of-the-art federated learning methods.

---

### Official Review · Reviewer_Lvm5 · 2024-07-15

**Soundness:** 3
**Presentation:** 3
**Contribution:** 2
**Rating:** 5
**Confidence:** 3

**Summary:**

This paper introduces Fed-POE, a novel personalized federated learning algorithm tailored for online prediction and model fine-tuning. Fed-POE creates an ensemble by integrating local models with those periodically contributed by the server over time. Theoretical analysis confirms that Fed-POE attains sublinear regret. Empirical results demonstrate that Fed-POE consistently surpasses the performance of both local and federated models across all evaluated datasets, which indicates that Fed-POE effectively leverages the advantages of both local and federated models.

**Strengths:**

- The technical content of the paper appears to be accurate, although I did not check all the details carefully.
- This paper is generally well-written and structured clearly.
- The experiments substantiate the main theoretical analysis, and the proposed algorithm demonstrates superior performance over the baseline methods

**Weaknesses:**

My primary concern is that the assertion the proposed algorithm can effectively harness the combined advantages of federated and local models is not clearly demonstrated within the theoretical bounds. The paper presents two principal theoretical results: Theorem 2 provides the regret upper bound for the proposed algorithm in convex scenarios, while Theorem 3 addresses non-convex cases. Both theorems establish sublinear regret bounds that are consistent with those for federated learning using a straightforward online gradient descent approach.  I recommend enhancing the clarity of the proposed method's advantages in the theorems by incorporating assumptions about the data distributions.

**Questions:**

See weaknesses.

**Limitations:**

Yes.

---

> ### Author Rebuttal · Authors · 2024-08-07
>
> Thank you very much for taking the time to review our paper and providing your valuable comments. Please find below our response to your review.
>
> The main advantage of the proposed Fed-POE compared to the straightforward online gradient descent approach is its ability to provide sublinear regret upper bounds for both global and personalized regret. The conventional method, as presented in Theorem 1, cannot guarantee a sublinear regret upper bound for personalized regret. The conventional online gradient descent approach only guarantees global regret. Fed-POE achieves sublinear regret upper bounds for personalized regret, as shown in Equations (12) and (20). Moreover, global regret upper bounds for Fed-POE are presented in Equations (11) and (19).

---

### Decision · Program_Chairs · 2024-09-25

**Decision:**

Accept (poster)

**Comment:**

This paper proposes a personalized federated learning algorithm, Fed-POE, which is designed for adaptive prediction and model fine-tuning in dynamic environments, based on the provocative claim that federated learning is not always better than locally-trained models. It is established on some sound theoretical analysis, and also evaluated to show robust performance in experiments. In the final version, we hope that the authors can improve the presentation as suggested by some reviewers, and incorporate more helpful details in the rebuttal into the main paper.